# Regret Minimization via Saddle Point Optimization

**Johannes Kirschner**
Department of Computer Science
University of Alberta
jkirschn@ualberta.ca

**Seyed Alireza Bakhtiari**
Department of Computer Science
University of Alberta
sbakhtia@ualberta.ca

**Kushagra Chandak**
Department of Computer Science
University of Alberta
kchandak@ualberta.ca

**Volodymyr Tkachuk**
Department of Computer Science
University of Alberta
vtkachuk@ualberta.ca

**Csaba Szepesvári**
Department of Computer Science
University of Alberta
szepesva@ualberta.ca

## Abstract

A long line of works characterizes the sample complexity of regret minimization in sequential decision-making by min-max programs. In the corresponding saddle-point game, the min-player optimizes the sampling distribution against an adversarial max-player that chooses confusing models leading to large regret. The most recent instantiation of this idea is the decision-estimation coefficient (DEC), which was shown to provide nearly tight lower and upper bounds on the worst-case expected regret in structured bandits and reinforcement learning. By reparametrizing the offset DEC with the confidence radius and solving the corresponding min-max program, we derive an anytime variant of the Estimation-To-Decisions algorithm (ANYTIME-E2D). Importantly, the algorithm optimizes the exploration-exploitation trade-off online instead of via the analysis. Our formulation leads to a practical algorithm for finite model classes and linear feedback models. We further point out connections to the information ratio, decoupling coefficient and PAC-DEC, and numerically evaluate the performance of E2D on simple examples.

## 1 Introduction

Regret minimization is a widely studied objective in bandits and reinforcement learning theory [Lattimore and Szepesvári, 2020a] that has inspired practical algorithms, for example, in noisy zero-order optimization[e.g., Srinivas et al., 2010] and deep reinforcement learning [e.g., Osband et al., 2016]. Cumulative regret measures the online performance of the algorithm by the total loss suffered due to choosing suboptimal decisions. Regret is unavoidable to a certain extent as the learner collects information to reduce uncertainty about the environment. In other words, a learner will inevitably face the exploration-exploitation trade-off where it must balance collecting rewards and collecting information. Finding the right balance is the central challenge of sequential decision-making under uncertainty.

More formally, denote by $\Pi$ a decision space and $\mathcal{O}$ an observation space. Let $\mathcal{M}$ be a class of models, where $f = (r_f, M_f) \in \mathcal{M}$ associated with a reward function $r_f : \Pi \to \mathbb{R}$ and observation

37th Conference on Neural Information Processing Systems (NeurIPS 2023).

map $M_f : \Pi \to \mathscr{P}(\mathcal{O})$, where $\mathscr{P}(\mathcal{O})$ is the set of all probability distributions over $\mathcal{O}$.[1] The learner's objective is to collect as much reward as possible in $n$ steps when facing a model $f^* \in \mathcal{M}$. The learner's prior information is $\mathcal{M}$ and the associated reward and observation maps, but does not know the true instance $f^* \in \mathcal{M}$. The learner constructs a stochastic sequence $\pi_1, \ldots, \pi_n$ of decisions taking values in $\Pi$ and adapted to the history of observations $y_t \sim M_{f^*}(\pi_t)$. The policy of the learner is the sequence of probability kernels $\mu_{1:n} = (\mu_t)_{t=1}^n$ that are used to take decisions. The expected regret of a policy $\mu_{1:n}$ and model $f^*$ after $n \in \mathbb{N}$ steps is

$$R_n(\mu_{1:n}, f^*) = \max_{\pi \in \Pi} \mathbb{E}\left[ \sum_{t=1}^n r_{f^*}(\pi) - r_{f^*}(\pi_t) \right]$$

The literature studies regret minimization for various objectives, including worst-case and instance-dependent frequentist regret [Lattimore and Szepesvári, 2020a], Bayesian regret [Russo and Van Roy, 2014] and robust variants [Garcelon et al., 2020, Kirschner et al., 2020a]. For the frequentist analysis, all prior knowledge is encoded in the model class $\mathcal{M}$. The worst-case regret of policy $\mu_{1:n}$ on $\mathcal{M}$ is $\sup_{f \in \mathcal{M}} R_n(\mu_{1:n}, f)$, and therefore the optimal minimax regret $\inf_\mu \sup_{f \in \mathcal{M}} R_n(\mu_{1:n}, f)$ only depends on $\mathcal{M}$ and the horizon $n$. The Bayesian, in addition, assumes access to a prior $\nu \in \mathscr{P}(\mathcal{M})$, which leads to the Bayesian regret $\mathbb{E}_{f \sim \nu}[R_n(\mu_{1:n}, f)]$. Interestingly, the worst-case frequentist regret and Bayesian regret are dual in the following sense [Lattimore and Szepesvári, 2019]:[2]

$$\inf_{\mu_{1:n}} \sup_{f \in \mathcal{M}} R_n(\mu_{1:n}, f) = \sup_{\nu \in \mathscr{P}(\mathcal{M})} \inf_{\mu_{1:n}} \mathbb{E}_{f \sim \nu}[R_n(\mu_{1:n}, f)] \tag{1}$$

Unfortunately, directly solving for the minimax policy (or the worst-case prior) is intractable, except in superficially simple problems. Ths is because the optimization is over the exponentially large space of adaptive policies. However, the relationship in Eq. (1) has been directly exploited in prior works, for example, to derive non-constructive upper bounds on the worst-case regret via a Bayesian analysis [Bubeck et al., 2015]. Moreover, it can be seen as inspiration underlying "optimization-based" algorithms for regret minimization: The crucial step is to carefully relax the saddle point problem in a way that preserves the statistical complexity, but can be analyzed and computed more easily. This idea manifests in several closely related algorithms, including information-directed sampling [Russo and Van Roy, 2014, Kirschner and Krause, 2018], ExpByOpt [Lattimore and Szepesvári, 2020b, Lattimore and Gyorgy, 2021], and most recently, the Estimation-To-Decisions (E2D) framework [Foster et al., 2021, 2023]. These algorithms have in common that they optimize the information trade-off directly, which in structured settings leads to large improvements compared to standard optimistic exploration approaches and Thompson sampling. On the other hand, algorithms that directly optimize the information trade-off can be computationally more demanding and, consequently, are often not the first choice of practitioners. This is partly due to the literature primarily focusing on statistical aspects, leaving computational and practical considerations underexplored.

**Contributions**    Building on the results by Foster et al. [2021], we introduce the *average-constrained decision-estimation coefficient* ($\mathrm{dec}_\epsilon^{ac}$), a saddle-point objective that characterizes the frequentist worst-case regret in sequential decision-making with structured observations. Compared to the decision-estimation coefficient of [Foster et al., 2021], the $\mathrm{dec}_\epsilon^{ac}$ is parametrized via the confidence radius $\epsilon$, instead of the Lagrangian offset multiplier. This allows optimization of the information trade-off online by the algorithm, instead of via the derived regret upper bound. Moreover, optimizing the $\mathrm{dec}_\epsilon^{ac}$ leads to an anytime version of the E2D algorithm (ANYTIME-E2D) with a straightforward analysis. We also point out relations between the $\mathrm{dec}_\epsilon^{ac}$, the information ratio [Russo and Van Roy, 2016], the decoupling coefficient [Zhang, 2022] and a PAC version of the DEC [Foster et al., 2023]. We further detail how to implement the algorithm for finite model classes and linear feedback models, and demonstrate the advantage of the approach by providing improved bounds for linear bandits with side-observations. Lastly, we report the first empirical results of the E2D algorithm on simple examples.

## 1.1   Related Work

There is a broad literature on regret minimization in bandits [Lattimore and Szepesvári, 2020a] and reinforcement learning [Jin et al., 2018, Azar et al., 2017, Zhou et al., 2021, Du et al., 2021,

---

[1]To simplify the presentation, we ignore tedious measure-theoretic details in this paper. The reader could either fill out the missing details, or just assume that all sets, unless otherwise stated, are discrete.

[2]The result by Lattimore and Szepesvári [2019] was only shown for finite action, reward and observation spaces, but can likely be extended to the infinite case under suitable continuity assumptions.

Zanette et al., 2020]. Arguably the most popular approaches are based on optimism, leading to the widely analysed upper confidence bound (UCB) algorithms [Lattimore and Szepesvári, 2020a], and Thompson sampling (TS) [Thompson, 1933, Russo and Van Roy, 2016].

A long line of work approaches regret minimization as a saddle point problem. Degenne et al. [2020b] showed that in the structured bandit setting, an algorithm based on solving a saddle point equation achieves asymptotically optimal regret bounds, while explicitly controlling the finite-order terms. Lattimore and Szepesvári [2020b] propose an algorithm based on exponential weights in the partial monitoring setting [Rustichini, 1999] that finds a distribution for exploration by solving a saddle-point problem. The saddle-point problem balances the trade-off between the exponential weights distribution and an information or stability term. The same approach was further refined by Lattimore and Gyorgy [2021]. In stochastic linear bandits, Kirschner et al. [2021] demonstrated that information-directed sampling can be understood as a primal-dual method solving the asymptotic lower bound, which leads to an algorithm that is both worst-case and asymptotically optimal. The saddle-point approach has been further explored in the PAC setting [e.g., Degenne et al., 2020b,a].

Our work is closely related to recent work by Foster et al. [2021, 2023]. They consider *decision making with structured observations* (DMSO), which generalizes the bandit and RL setting. They introduce a complexity measure, the *offset decision-estimation coefficient* (offset DEC), defined as a min-max game between a learner and an environment, and provide lower bounds in terms of the offset DEC. Further, they provide an algorithm, *Estimation-to-Decisions* (E2D) with corresponding worst-case upper bounds in terms of the offset DEC. Notably, the lower and upper bound nearly match and recover many known results in bandits and RL. More recently, Foster et al. [2023] refined the previous bounds by introducing the *constrained* DEC and a corresponding algorithm E2D$^+$.

There are various other results related to the DEC and the E2D algorithm. Foster et al. [2022a] show that the E2D achieves improved bounds in model-free RL when combined with optimistic estimation (as introduced by Zhang [2022]). Chen et al. [2022] introduced two new complexity measures based on the DEC that are necessary and sufficient for reward-free learning and PAC learning. They also introduced new algorithms based on the E2D algorithm for the above two settings and various other improvements. Foster et al. [2022b] have shown that the DEC is necessary and sufficient to obtain low regret for *adversarial* decision-making. An asymptotically instance-optimal algorithm for DMSO has been proposed by Dong and Ma [2022], extending a similar approach for the linear bandit setting [Lattimore and Szepesvari, 2017].

The decision-estimation coefficient is also related to the information ratio [Russo and Van Roy, 2014] and the decoupling coefficient [Zhang, 2022]. The information ratio has been studied under both the Bayesian [Russo and Van Roy, 2014] and the frequentist regret [Kirschner and Krause, 2018, Kirschner et al., 2020b, 2021, 2023] in various settings including bandits, reinforcement learning, and partial monitoring. The decoupling coefficient was studied for the Thompson sampling algorithm in contextual bandits [Zhang, 2022], and RL [Dann et al., 2021, Agarwal and Zhang, 2022].

## 2  Setting

We consider the sequential decision-making problem already introduced in the preface. Recall that $\Pi$ is a compact decision space and $\mathcal{O}$ is an observation space. The model class $\mathcal{M}$ is a set of tuples $f = (r_f, M_f)$ containing a reward function $r_f : \Pi \to \mathbb{R}$ and an observation distribution $M_f : \Pi \to \mathscr{P}(\mathcal{O})$. We define the gap function

$$\Delta(\pi, g) = r_g(\pi_g^*) - r_g(\pi),$$

where $\pi_g^* = \arg\max_{\pi \in \Pi} r_g(\pi)$ is an optimal decision for model $g$, chosen arbitrarily if not unique. A randomized policy is a sequence of kernels $\mu_{1:n} = (\mu_t)_{t=1}^n$ from histories $h_{t-1} = (\pi_1, y_1, \ldots, \pi_{t-1}, y_{t-1}) \in (\Pi \times \mathcal{O})^{t-1}$ to sampling distributions $\mathscr{P}(\Pi)$. The filtration generated by the history $h_t$ is $\mathcal{F}_t$. The learner's decisions $\pi_1, \ldots, \pi_n$ are sampled from the policy $\pi_t \sim \mu_t$ and observations $y_t \sim M_{f^*}(\pi_t)$ are generated by an unknown true model $f^* \in \mathcal{M}$. The expected regret under model $f^*$ is formally defined as follows:

$$R_n(\mu_{1:n}, f^*) = \mathbb{E}\left[\sum_{t=1}^n \mathbb{E}_{\pi_t \sim \mu_t(h_t)}[\Delta(\pi_t, f^*)]\right]$$

For now, we do not make any assumption about the reward being observed. This provides additional flexibility to model a wide range of scenarios, including for example, duelling and ranking feedback [Yue and Joachims, 2009, Radlinski et al., 2008, Combes et al., 2015, Lattimore et al., 2018, Kirschner and Krause, 2021] (e.g. used in reinforcement learning with human feedback, RLHF) or dynamic pricing [den Boer, 2015]. The setting is more widely known as partial monitoring Rustichini [1999]. The special case where the reward is part of the observation distribution is called *decision-making with structured observations* [DMSO, Foster et al., 2021]. Earlier work studies the closely related *structured bandit* setting [Combes et al., 2017].

A variety of examples across bandit models and reinforcement learning are discussed in [Combes et al., 2017, Foster et al., 2021, 2023, Kirschner et al., 2023]. For the purpose of this paper, we focus on simple cases for which we can provide tractable implementations. Besides the finite setting where $\mathcal{M}$ can be enumerated, these are the following linearly parametrized feedback models.

**Example 2.1** (Linear Bandits, Abe and Long [1999]). The model class is identified with a subset of $\mathbb{R}^d$ and features $\phi_\pi \in \mathbb{R}^d$ for each $\pi \in \Pi$. The reward function is $r_f(\pi) = \langle \phi(\pi), f \rangle$ and the observation distribution is $M_f(\pi) = \mathcal{N}(\langle \phi_\pi, f \rangle, 1)$.

The linear bandit setting can be generalized by separating reward and feedback maps, which leads to the *linear partial monitoring* framework [Lin et al., 2014, Kirschner et al., 2020b]. Here we restrict our attention to the special case of *linear bandits with side-observations* [c.f. Kirschner et al., 2023], which, for example, generalizes the classical semi-bandit setting Mannor and Shamir [2011]

**Example 2.2** (Linear Bandits with Side-Observations). As in the linear bandit setting, we have $\mathcal{M} \subset \mathbb{R}^d$, and features $\phi_\pi \in \mathbb{R}^d$ that define the reward functions $r_f(\pi) = \langle \phi_\pi, f \rangle$. Observation matrices $M_\pi \in \mathbb{R}^{m_\pi \times d}$ for each $\pi \in \Pi$ define $m_\pi$-dimensional observation distributions $M_f(\pi) = \mathcal{N}(M_\pi f, \sigma^2 \mathbf{1}_{m_\pi})$. In addition, we assume that $\phi_\pi \phi_\pi^\top \preceq M_\pi^\top M_\pi$, which is automatically satisfied if $\phi_\pi^\top$ is included in the rows of $M_\pi$, i.e. when the reward is part of the observations.

## 3 Regret Minimization via Saddle-Point Optimization

The goal of the learner is to choose decisions $\pi \in \Pi$ that achieve a small gap $\Delta(\pi, f^*)$ under the true model $f^* \in \mathcal{M}$. Since the true model is unknown, the learner has to collect data that provides statistical evidence to reject models $g \neq f^*$ for which the regret $\Delta(\pi, g)$ is large. To quantify the information-regret trade-off, we use a divergence $D(\cdot \| \cdot)$ defined for distributions in $\mathscr{P}(\mathcal{O})$. For a reference model $f$, the information (or divergence) function is defined by:

$$I_f(\pi, g) = D_{\mathrm{KL}}(M_g(\pi) \| M_f(\pi)),$$

where $D_{\mathrm{KL}}(\cdot \| \cdot)$ is the KL divergence. Intuitively, $I_f(\pi, g)$ is the rate at which the learner collects statistical information to reject $g \in \mathcal{M}$ when choosing $\pi \in \Pi$ and data is generated under the reference model $f$. Note that $I_f(\pi, f) = 0$ for all $f \in \mathcal{M}$ and $\pi \in \Pi$. As we will see shortly, the regret-information trade-off can be written precisely as a combination of the gap function, $\Delta$, and the information function, $I_f$. We remark in passing that other choices such as the Hellinger distance are also possible, and the KL divergence is mostly for concreteness and practical reasons.

To simplify the notation and emphasize the bilinear nature of the saddle point problem that we study, we will view $\Delta, I_f \in \mathbb{R}_+^{\Pi \times \mathcal{M}}$ as $|\Pi| \times |\mathcal{M}|$ matrices (by fixing a canonical ordering on $\Pi$ and $\mathcal{M}$). For vectors $\mu \in \mathbb{R}^\Pi$ and $\nu \in \mathbb{R}^\mathcal{M}$, we will frequently write bilinear forms $\mu \Delta_f \nu$ and $\mu I_f \nu$. This also means that by convention, $\mu$ will always denote a row vector, while $\nu$ will always denote a column vector. The standard basis for $\mathbb{R}^\Pi$ and $\mathbb{R}^\mathcal{M}$ is $(e_\pi)_{\pi \in \Pi}$ and $(e_g)_{g \in \mathcal{M}}$.

### 3.1 The Decision-Estimation Coefficient

To motivate our approach, we recall the *decision-estimation coefficient* (DEC) introduced by Foster et al. [2021, 2023], before introducing the main quantity of interest, the *average-constrained DEC*. First, the *offset decision-estimation coefficient* (without localization) [Foster et al., 2021] is

$$\mathrm{dec}_\lambda^o(f) = \min_{\mu \in \mathscr{P}(\Pi)} \max_{g \in \mathcal{M}} \mu \Delta e_g - \lambda \mu I_f e_g$$

The tuning parameter $\lambda > 0$ controls the weight of the information matrix relative to the gaps: Viewing the above as a two-player zero-sum game, we see that increasing $\lambda$ forces the max-player

to avoid models that differ significantly from $f$ under the min-player's sampling distribution. The advantage of this formulation is that the information term $\mu I_f e_g$ can be telescoped in the analysis, which directly leads to regret bounds in terms of the estimation error (introduced below in Eq. (8)). The disadvantage of the $\lambda$-parametrization is that the trade-off parameter is chosen by optimizing the final regret upper bound. This is inconvenient because the optimal choice requires knowledge of the horizon and a bound on $\max_{f \in \mathcal{M}} \text{dec}_\lambda^o(f)$. Moreover, any choice informed by the upper bound may be conservative, leading to sub-optimal performance.

The *constrained decision-estimation coefficient* [Foster et al., 2023] is

$$\text{dec}_\epsilon^c(f) = \min_{\mu \in \mathscr{P}(\Pi)} \max_{g \in \mathcal{M}} \mu \Delta e_g \qquad \text{s.t.} \qquad \mu I_f e_g \leq \epsilon^2 \tag{2}$$

In this formulation, the max player is restricted to choose models $g$ that differ from $f$ at most by $\epsilon^2$ in terms of the observed divergence under the min-player's sampling distribution. Note that because $e_\pi I_f e_f = 0$ for all $e_\pi \in \Pi$, there always exists a feasible solution. For horizon $n$, the radius can be set to $\epsilon^2 \approx \frac{\beta_\mathcal{M}}{n}$, where $\beta_\mathcal{M}$ is a model estimation complexity parameter, thereby essentially eliminating the trade-off parameter from the algorithm. However, because of the hard constraint, strong duality of the Lagrangian saddle point problem (for fixed $\mu$) fails, and consequently, telescoping the information gain in the analysis is no longer easily possible (or at least, with the existing analysis). To achieve sample complexity $\text{dec}_\epsilon^c(f)$, Foster et al. [2023] propose a sophisticated scheme that combines phased exploration with a refinement procedure (E2D$^+$).

As the main quantity of interest in the current work, we now introduce the *average-constrained decision-estimation coefficient*, defined as follows:

$$\text{dec}_\epsilon^{ac}(f) = \min_{\mu \in \mathscr{P}(\Pi)} \max_{\nu \in \mathscr{P}(\mathcal{M})} \mu \Delta \nu \qquad \text{s.t.} \qquad \mu I_f \nu \leq \epsilon^2 \tag{3}$$

Similar to the $\text{dec}_\epsilon^c$, the parameterization of the $\text{dec}_\epsilon^{ac}$ is via the confidence radius $\epsilon^2$, making the choice of the hyperparameter straightforward in many cases (more details in Section 3.2). By convexifying the domain $\mathscr{P}(\mathcal{M})$ of the max-player, we recover strong duality of the Lagrangian (for fixed $\mu$). Thereby, the formulation inherits the ease of choosing the $\epsilon$-parameter from the $\text{dec}_\epsilon^c$, while, at the same time, admitting a telescoping argument in the analysis and a much simpler algorithm.

Specifically, Sion's theorem implies three equivalent Lagrangian representations for Eq. (3):

$$\text{dec}_\epsilon^{ac}(f) = \min_{\mu \in \mathscr{P}(\Pi)} \max_{\nu \in \mathscr{P}(\mathcal{M})} \min_{\lambda \geq 0} \mu \Delta \nu - \lambda(\mu I_f \nu - \epsilon^2) \tag{4}$$

$$= \min_{\lambda \geq 0, \mu \in \mathscr{P}(\Pi)} \max_{\nu \in \mathscr{P}(\mathcal{M})} \mu \Delta \nu - \lambda(\mu I_f \nu - \epsilon^2) \tag{5}$$

$$= \min_{\lambda \geq 0} \max_{\nu \in \mathscr{P}(\mathcal{M})} \min_{\mu \in \mathscr{P}(\Pi)} \mu \Delta \nu - \lambda(\mu I_f \nu - \epsilon^2) \tag{6}$$

When fixing the outer problem, strong duality holds for the inner saddle-point problem in each line, however, the joint program in Eq. (5) is not convex-concave. An immediate consequence of relaxing the domain of the max player and Eq. (5) is that

$$\text{dec}_\epsilon^c(f) \leq \text{dec}_\epsilon^{ac}(f) = \min_{\lambda \geq 0}\{\text{dec}_\lambda^o(f) + \lambda \epsilon^2\} \tag{7}$$

The $\text{dec}_\epsilon^{ac}$ can therefore be understood as setting the $\lambda$ parameter of the $\text{dec}_\lambda^o$ optimally for the given confidence radius $\epsilon^2$. On the other hand, the cost paid for relaxing the program is that there exist model classes $\mathcal{M}$ where the inequality in Eq. (7) is strict, and $\text{dec}_\epsilon^{ac}$ does not lead to a tight characterization of the regret [Foster et al., 2023, Proposition 4.4]. The remedy is that under a stronger regularity condition and localization, the two notions are essentially equivalent [Foster et al., 2023, Proposition 4.8].

## 3.2 Anytime Estimation-To-Decisions (Anytime-E2D)

Estimations-To-Decisions (E2D) is an algorithmic framework that directly leverages the decision-estimation coefficient for choosing a decision in each round. The key idea is to compute a sampling distribution $\mu_t \in \mathscr{P}(\Pi)$ attaining the minimal DEC for an estimate $\hat{f}_t$ of the underlying model, and then define the policy to sample $\pi_t \sim \mu_t$. The E2D approach, using the $\text{dec}_\epsilon^{ac}$ formulation, is summarized in Algorithm 1. To compute the estimate $\hat{f}_t$, the E2D algorithm takes an abstract estimation oracle EST as input, that, given the collected data, returns $\hat{f}_t \in \mathcal{M}$. The final guarantee

---

**Algorithm 1:** ANYTIME-E2D

---

**Input :** Hypothesis class $\mathcal{M}$, estimation oracle EST, sequence $\epsilon_t \geq 0$, data $\mathcal{D}_0 = \emptyset$

1 **for** $t = 1, 2, 3, \ldots$ **do**
2     Estimate $\hat{f}_t = \text{EST}(\mathcal{D}_{t-1})$
3     Compute gap and information matrices, $\Delta$ and $I_{\hat{f}_t} \in \mathbb{R}^{\Pi \times \mathcal{M}}$
4     $\mu_t = \arg\min_{\mu \in \mathscr{P}(\Pi)} \max_{\nu \in \mathscr{P}(\mathcal{M})} \{\mu\Delta\nu : \mu I_{\hat{f}_t}\nu \leq \epsilon_t^2\}$
5     Sample $\pi_t \sim \mu_t$ and observe $y_t \sim M_{f^*}(\pi_t)$
6     Append data $\mathcal{D}_t = \mathcal{D}_{t-1} \cup \{(\pi_t, y_t)\}$

---

depends on the *estimation error* (or estimation regret), defined as the sum over divergences of the observation distributions under the estimate $\hat{f}_t$ and the true model $f^*$:

$$\text{Est}_n = \mathbb{E}\left[\sum_{t=1}^{n} \mu_t I_{\hat{f}_t} e_{f^*}\right] \tag{8}$$

Intuitively, the estimation error is well-behaved if $\hat{f}_t \approx f^*$, since $\mu_t I_{f^*} e_{f^*} = 0$. Equation (8) is closely related to the *total information gain* used in the literature on information-directed sampling [Russo and Van Roy, 2014] and kernel bandits [Srinivas et al., 2010].

To bound the estimation error, Foster et al. [2021] rely on *online density estimation* (also, *online regression* or *online aggregation*) [Cesa-Bianchi and Lugosi, 2006, Chapter 9]. For finite $\mathcal{M}$, the default approach is the *exponential weights algorithm* (EWA), which we provide for reference in Appendix A. When using this algorithm, the estimation error always satisfies $\text{Est}_n \leq \log(|\mathcal{M}|)$, see [Cesa-Bianchi and Lugosi, 2006, Proposition 3.1]. While these bounds extend to continuous model classes via standard covering arguments, the resulting algorithm is often not tractable without additional assumptions. For linear feedback models (Examples 2.1 and 2.2), one can rely on the more familiar ridge regression estimator, which, we show, achieves bounded estimation regret $\text{Est}_n \leq \mathcal{O}(d\log(n))$. For further discussion, see Appendix A.1.

With this in mind, we state our main result.

**Theorem 1.** *Let $\lambda_t \geq 0$ be any sequence adapted to the filtration $\mathcal{F}_t$. Then the regret of* ANYTIME-E2D *(Algorithm 1) with input sequence $\lambda_t$ satisfies for all $n \geq 1$:*

$$R_n \leq \underset{t \in [n]}{\text{ess sup}}\left\{\frac{\text{dec}_{\epsilon_t, \lambda_t}^{ac}(\hat{f}_t)}{\epsilon_t^2}\right\}\left(\sum_{t=1}^{n} \epsilon_t^2 + \text{Est}_n\right)$$

*where we defined* $\text{dec}_{\epsilon, \lambda}^{ac}(f) = \min_{\mu \in \mathscr{P}(\Pi)} \max_{\nu \in \mathscr{P}(\mathcal{M})} \mu\Delta\nu - \lambda(\mu I_f \nu - \epsilon^2)$.

As an immediate corollary, we obtain a regret bound for Algorithm 1 where the sampling distribution $\mu_t$ is chosen to optimize $\text{dec}_{\epsilon_t}^{ac}$ for any sequence $\epsilon_t$.

**Corollary 1.** *The regret of* ANYTIME-E2D *(Algorithm 1) with input $\epsilon_t \geq 0$ satisfies for all $n \geq 1$:*

$$R_n \leq \max_{t \in [n], f \in \mathcal{M}}\left\{\frac{\text{dec}_{\epsilon_t}^{ac}(f)}{\epsilon_t^2}\right\}\left(\sum_{t=1}^{n} \epsilon_t^2 + \text{Est}_n\right)$$

Importantly, the regret of Algorithm 1 is directly controlled by the worst-case DEC, $\max_{f \in \mathcal{M}} \text{dec}_\epsilon^{ac}(f)$, and the estimation error $\text{Est}_n$. It remains to set $\epsilon_t^2$ (respectively $\lambda_t$) appropriately. For a fixed horizon $n$, we let $\epsilon_t^2 = \frac{\text{Est}_n}{n}$. With the reasonable assumption that $\max_{f \in \mathcal{M}}\{\epsilon^{-2}\text{dec}_\epsilon^{ac}(f)\}$ is non-decreasing in $\epsilon$, Corollary 1 reads

$$R_n \leq 2n \max_{f \in \mathcal{M}}\left\{\text{dec}_{\sqrt{\text{Est}_n/n}}^{ac}(f)\right\}. \tag{9}$$

This almost matches the lower bound $R_n \geq \Omega(n\text{dec}_{1/\sqrt{n}}^{c}(\mathcal{F}))$[3] [Foster et al., 2023, Theorem 2.2], up to the estimation error and the beforehand mentioned gap between $\text{dec}_\epsilon^{c}$ and $\text{dec}_\epsilon^{ac}$.

---

[3]Here, $\text{dec}_\epsilon^{c}(\mathcal{F}) = \max_{f \in \text{co}(\mathcal{M})} \min_{\mu \in \mathscr{P}(\Pi)} \max_{g \in \mathcal{M} \cup \{f\}} \{\mu\Delta\nu : \mu I_f e_g \leq \epsilon^2\}$.

| Setting | $\text{dec}_\gamma^o$ | $\text{dec}_\epsilon^{ac}$ |
|---|---|---|
| Multi-Armed Bandits | $|\Pi|/\gamma$ | $2\epsilon\sqrt{|\Pi|}$ |
| Linear Bandits | $d/4\gamma$ | $\epsilon\sqrt{d}$ |
| Lipschitz Bandits | $2\gamma^{-\frac{1}{d+1}}$ | $2^{\frac{d+1}{d+2}}\epsilon^{\frac{2}{d+2}}$ |
| Convex Bandits | $\tilde{O}(d^4/\gamma)$ | $\tilde{O}(\epsilon d^2)$ |

Table 1: Comparison of $\text{dec}_\gamma^o$ and $\text{dec}_\epsilon^{ac}$ for different settings. Bounds between $\text{dec}_\gamma^o$ and $\text{dec}_\epsilon^{ac}$ can be converted using Eq. (7). Refined bounds for linear bandits with side-observations are in Lemma 4.

To get an anytime algorithm with essentially the same scaling as in Eq. (9), we set $\epsilon_t^2 = \log(|\mathcal{M}|)/t$ for finite model classes, and $\epsilon_t^2 = \frac{\beta_\mathcal{M}}{t}$ if $\text{Est}_t \leq \beta_\mathcal{M}\log(t)$ for $\beta_\mathcal{M} > 0$. For linear bandits, $\text{dec}_\epsilon^{ac} \leq \epsilon\sqrt{d}$ (see Section 3.3), and $\text{Est}_n \leq d\log(n)$. Choosing $\epsilon_t^2 = d/t$ recovers the optimal regret bound $R_n \leq \tilde{\mathcal{O}}(d\sqrt{n})$ [Lattimore and Szepesvári, 2020a]. Alternatively, one can also choose $\lambda_t$ by minimizing an upper bound on $\max_{t\in[n],f\in\mathcal{M}}\{\text{dec}_{\epsilon_t,\lambda_t}^{ac}(f)/\epsilon_t^2\}$. For example, in linear bandits, $\text{dec}_{\epsilon_t,\lambda}^{ac} \leq \frac{d}{4\lambda} + \lambda\epsilon_t^2$ (see Table 1); hence, for $\epsilon_t^2 = d/t$, we can set $\lambda_t = t/4$. Further discussion and refined upper bound for linear feedback models are in Section 3.3.

*Proof of Theorem 1.* Let $\mu_t^*$ and $\nu_t^*$ be a saddle-point solution to the offset dec,

$$\text{dec}_{\lambda_t}^o(\hat{f}_t) = \min_{\mu\in\mathscr{P}(\Pi)}\max_{\nu\in\mathscr{P}(\mathcal{M})}\mu\Delta\nu - \lambda_t\mu I_{\hat{f}_t}\nu$$

Note that $\mu_t^*\Delta\nu_t^* - \lambda_t\mu_t^* I_f\nu_t^* \geq \mu_t^*\Delta e_f - \lambda_t\mu_t^* I_f e_f \geq 0$, which implies that $\lambda_t\epsilon_t^2 \leq \text{dec}_{\epsilon_t,\lambda_t}^{ac}$. Next,

$$R_n = \mathbb{E}\left[\sum_{t=1}^n \mu_t\Delta e_{f^*}\right] = \sum_{t=1}^n \mathbb{E}\left[\mu_t\Delta e_{f^*} - \lambda_t(\mu_t I_{\hat{f}_t}e_{f^*} - \epsilon_t^2) + \lambda_t(\mu_t I_{\hat{f}_t}e_{f^*} - \epsilon_t^2)\right]$$

$$\leq \sum_{t=1}^n \mathbb{E}\left[\max_{g\in\mathcal{M}}\mu_t\Delta_{\hat{f}_t}e_g - \lambda_t(\mu_t I_{\hat{f}_t}e_g - \epsilon_t^2) + \lambda_t(\mu_t I_{\hat{f}_t}e_{f^*} - \epsilon_t^2)\right]$$

$$= \sum_{t=1}^n \mathbb{E}\left[\min_{\mu\in\mathscr{P}(\Pi)}\max_{\nu\in\mathscr{P}(\mathcal{M})}\mu\Delta\nu - \lambda_t(\mu I_{\hat{f}_t}\nu - \epsilon_t^2) + \lambda_t(\mu_t I_{\hat{f}_t}e_{f^*} - \epsilon_t^2)\right]$$

So far, we only introduced the saddle point problem by maximizing over $f^*$. The last equality is by our choice of $\lambda_t$ and $\mu_t$, and noting that $\nu \in \mathscr{P}(\mathcal{M})$ can always be realized as a Dirac. Continuing,

$$R_n \leq \sum_{t=1}^n \mathbb{E}\left[\text{dec}_{\epsilon_t,\lambda_t}^{ac}(\hat{f}_t) + \lambda_t(\mu_t I_{\hat{f}_t}e_{f^*} - \epsilon_t^2)\right]$$

$$\stackrel{(i)}{\leq} \sum_{t=1}^n \mathbb{E}\left[\text{dec}_{\epsilon_t,\lambda_t}^{ac}(\hat{f}_t) + \frac{1}{\epsilon_t^2}\text{dec}_{\epsilon_t,\lambda_t}^{ac}(\hat{f}_t)\mu_t I_{\hat{f}_t}e_{f^*}\right]$$

$$\stackrel{(ii)}{\leq} \underset{t\in[n]}{\text{ess sup}}\max_{f\in\mathcal{M}}\left\{\frac{1}{\epsilon_t^2}\text{dec}_{\epsilon_t,\lambda_t}^{ac}(f)\right\}\sum_{t=1}^n\left(\epsilon_t^2 + \mathbb{E}\left[\mu_t I_{\hat{f}_t}e_{f^*}\right]\right)$$

We first drop the negative term in $(i)$ and use the beforehand stated fact that $\lambda_t\epsilon_t^2 \leq \text{dec}_{\epsilon_t,\lambda_t}^{ac}(\hat{f}_t)$. The last step, $(ii)$, is taking the maximum out of the sum. $\qquad\square$

## 3.3 Certifying Upper Bounds

As shown by Corollary 1, the regret of Algorithm 1 scales directly with the $\text{dec}_\epsilon^{ac}$. For analysis purposes, it is however useful to compute upper bounds on the $\text{dec}_\epsilon^{ac}$ to verify the scaling w.r.t. parameters of interest. Via the equivalence Eq. (7), bounds on the $\text{dec}_\lambda^o$ directly translate to the $\text{dec}_\epsilon^{ac}$ (see Table 1). For a detailed discussion of upper bounds in various models, we refer to Foster et al. [2021]. Below, we highlight three connections that are directly facilitated by the $\text{dec}_\epsilon^{ac}$.

To this end, we first introduce a variant of the $\mathrm{dec}_\epsilon^{ac}$ where the gap function depends on $f$:

$$\mathrm{dec}_\epsilon^{ac,f}(f) = \min_{\mu \in \mathscr{P}(\Pi)} \max_{\nu \in \mathscr{P}(\mathcal{M})} \mu \Delta_f \nu \qquad \text{s.t.} \qquad \mu I_f \nu \le \epsilon^2, \qquad (10)$$

where $\Delta_f(\pi, g) = r_g(\pi_g^*) - r_f(\pi)$. We remark that for distributions $\nu \in \mathscr{P}(\mathcal{M})$ and $\mu \in \mathscr{P}(\Pi)$, the gap $\Delta_f$ can be decoupled, $\mu \Delta_f \nu = \delta_f \nu + \mu \Delta_f e_f$, where we defined $\delta_f(g) = r_g(\pi_g^*) - r_f(\pi_f^*)$. The following assumption implies that the observations for a decision $\pi$ are at least as informative as observing the rewards.

**Assumption 1** (Reward Data Processing). *The rewards and information matrices are related via the following data-processing inequality that holds for any $\mu \in \mathscr{P}(\Pi)$:*

$$|\mathbb{E}_{\pi \sim \mu}[r_f(\pi) - r_g(\pi)]| \le \sqrt{\mathbb{E}_{\pi \sim \mu}[D(M_f(\pi) \| M_g(\pi))]}$$

The next lemma shows that under Assumption 1, $\mathrm{dec}_\epsilon^{ac}(f)$ and $\mathrm{dec}_\epsilon^{ac,f}(f)$ are essentially equivalent, at least for the typical worst-case bounds where $\max_{f \in \mathcal{M}} \mathrm{dec}_\epsilon^{ac}(f) \ge \Omega(\epsilon)$.

**Lemma 1.** *If Assumption 1 holds, then*

$$\mathrm{dec}_\epsilon^{ac,f}(f) - \epsilon \le \mathrm{dec}_\epsilon^{ac}(f) \le \mathrm{dec}_\epsilon^{ac,f}(f) + \epsilon$$

The proof is in Appendix C.1. We remark that Algorithm 1 where the sampling distribution is computed for $\mathrm{dec}_\epsilon^{ac,f}(\hat{f}_t)$ and $\Delta_f$ achieves a bound analogous to Theorem 1, as long as Assumption 1 holds. For details see Lemma 8 in Appendix C.

**Upper Bounds via Decoupling**   First, we introduce the *information ratio*,

$$\Psi_f(\mu, \nu) = \frac{(\mu \Delta_f \nu)^2}{\mu I_f \nu}$$

The definition is closely related to the Bayesian information ratio [Russo and Van Roy, 2016], where $\nu$ takes the role of a prior over $\mathcal{M}$. The Thompson sampling distribution is $\mu_\nu^{\mathrm{TS}} = \sum_{h \in \mathcal{M}} \nu_h e_{\pi_h^*}$. The decoupling coefficient, $\mathrm{dc}(f)$, [Zhang, 2022, Definition 1] is defined as the smallest number $K \ge 0$, such that for all distributions $\nu \in \mathscr{P}(\mathcal{M})$,

$$\mu_\nu^{\mathrm{TS}} \Delta_f \nu \le \inf_{\eta \ge 0} \left\{ \eta \sum_{g, h \in \mathcal{M}} \nu_g \nu_h e_{\pi_h^*} (r_g - r_f)^2 + \frac{K}{4\eta} \right\} = \sqrt{K \sum_{g, h \in \mathcal{M}} \nu_g \nu_h e_{\pi_h^*} (r_g - r_f)^2} \quad (11)$$

The next lemma provides upper bounds on the $\mathrm{dec}_\epsilon^{ac}(f)$ in terms of the information ratio, which is further upper-bounded by the decoupling coefficient.

**Lemma 2.** *With $\Psi(f) = \max_{\nu \in \mathcal{M}} \min_{\mu \in \mathscr{P}(\Pi)} \Psi_f(\mu, \nu)$ and Assumption 1 satisfied, we have*

$$\mathrm{dec}_\epsilon^{ac,f}(f) \le \epsilon \sqrt{\Psi(f)} \le \epsilon \sqrt{\mathrm{dc}(f)}$$

The proof follows directly using the AM-GM inequality, see Appendix C.2. By [Zhang, 2022, Lemma 2], this further implies $\mathrm{dec}_\epsilon^{ac,f} \le \epsilon \sqrt{d}$. An analogous result for the generalized information ratio [Lattimore and Gyorgy, 2021] that recovers rates $\epsilon^\rho$ for $\rho \le 1$ is given in Appendix C.4.

**PAC to Regret**   Another useful way to upper bound the $\mathrm{dec}_\epsilon^{ac,f}$ is via an analogous definition for the PAC setting [c.f. Eq. (10), Foster et al., 2023]:

$$\text{pac-dec}_\epsilon^{ac,f}(f) = \min_{\mu \in \mathscr{P}(\Pi)} \max_{\nu \in \mathcal{M}} \delta_f \nu \qquad \text{s.t.} \qquad \mu I_f \nu \le \epsilon^2 \qquad (12)$$

**Lemma 3.** *Under Assumption 1,*

$$\mathrm{dec}_\epsilon^{ac,f}(f) \le \min_{p \in [0,1]} \left\{ \text{pac-dec}_{\epsilon p^{-1/2}}^{ac,f}(f) + p \Delta_{\max} \right\}$$

The proof is given in Appendix B.1. Lemma 3 combined with Theorem 1 leads to $\mathcal{O}(n^{2/3})$ upper bounds on the regret that are reminiscent of so-called globally observable games in linear partial monitoring [Kirschner et al., 2023].

**Application to Linear Feedback Models** To illustrate the techniques introduced, we compute a regret bound for Algorithm 1 for linear bandits with side-observations (Examples 2.1 and 2.2).

**Lemma 4.** *For linear bandits with side-observations and divergence $I_f(\pi, g) = \|M_\pi(g - f)\|^2$,*

$$\text{pac-dec}_\epsilon^{ac,f}(f) \leq \min_{\mu \in \mathscr{P}(\Pi)} \max_{b \in \Pi} \epsilon \|\phi_b\|_{V(\mu)^{-1}} \leq \epsilon\sqrt{d}$$

*where $V(\mu) = \sum_{\pi \in \Pi} \mu_\pi M_\pi M_\pi^\top$. Moreover, denoting $\Omega = \min_{\mu \in \mathscr{P}(\Pi)} \max_{b \in \Pi} \|\phi_b\|_{V(\mu)^{-1}}$,*

$$\text{dec}_\epsilon^{ac,f}(f) \leq \min\left(\epsilon\sqrt{\Psi(f)}, 2\epsilon^{2/3}\Omega^{1/3}\Delta_{\max}^{1/3}\right)$$

The proof is given in Appendix B.2. While in the worst-case for linear bandits, there is no improvement over the standard $\mathcal{O}(d\sqrt{n})$ without further refinement or specification of the upper bounds, in the case of linear side-observations there is an improvement whenever $\Omega \leq \max_{f \in \mathcal{M}} \Psi(f)$. To exemplify the improvement, consider a semi-bandit with a "revealing" action $\hat{\pi}$, e.g. $M_{\hat{\pi}} = \mathbf{1}_d$. Here, the regret bound improves to $R_n \leq \min\{d\sqrt{n}, d^{1/3}n^{2/3}\}$, since then pac-dec$_\epsilon^{ac,f}(f) \leq \epsilon$. The corresponding improvement in the regime $n \leq d^4$ might seem modest, but is relevant in high-dimensional and non-parametric models. Moreover, in (deep) reinforcement learning, high-dimensional models are commonly used and the learner obtains side information in the form of state observations. Therefore, it is plausible that the $n^{2/3}$ rate is dominant even for a moderate horizon. Exploring this effect in reinforcement learning is therefore an important direction for future work.

Notably, this improvement is *not* observed by upper confidence bound algorithms and Thompson sampling, because both approaches discard informative but suboptimal actions early on [c.f. Lattimore and Szepesvari, 2017], including the action $\hat{\pi}$ in the example above. E2D for a constant offset parameter $\lambda > 0$, in principle, attains the better rate, but only if one pre-commits to a fixed horizon. Lastly, we note that a similar effect was observed for information-directed sampling in sparse high-dimensional linear bandits [Hao et al., 2020].

### 3.4 Computational Aspects

For finite model classes, Algorithm 1 can be readily implemented. Since almost no structure is imposed on the gap and information matrices of size $|\Pi| \times |\mathcal{M}|$, avoiding scaling with $|\Pi| \cdot |\mathcal{M}|$ seems hardly possible without introducing additional assumptions. Even in the finite case, solving Eq. (3) is not immediate because the corresponding Lagrangian is not convex-concave. A practical approach is to solve the inner saddle point for Eq. (5) as a function of $\lambda$. Strong duality holds for the inner problem, and one can obtain a solution efficiently by solving the corresponding linear program using standard solvers. It then remains to optimize over $\lambda \geq 0$. This can be done, for example, via a grid search over the range $[0, \max_{f \in \mathcal{M}} \epsilon^{-2}\text{dec}_\epsilon^{ac}(f)]$.

In the linear setting, the above is not satisfactory because most commonly $\mathcal{M}$ is identified with parameters in $\mathbb{R}^d$. As noted before, ridge regression can be used instead of online aggregation while preserving the optimal scaling of the estimation error (see Appendix A.1). The next lemma further shows that the saddle point problem Eq. (3) can be rewritten to only scale with the size of the decision set $|\Pi|$.

**Lemma 5.** *Consider linear bandits with side observations, $\mathcal{M} = \mathbb{R}^d$ and quadratic divergence, $I_f(\pi, g) = \|M_\pi(g - f)\|^2$, and denote $\phi_\mu = \sum_{\pi \in \Pi} \mu_\pi \phi_\pi$ and $V(\mu) = \sum_{\pi \in \Pi} \mu_\pi M_\pi^\top M_\pi$. Then*

$$\text{dec}_\epsilon^{ac,f}(\hat{f}_t) = \min_{\lambda \geq 0} \min_{\mu \in \mathscr{P}(\Pi)} \max_{b \in \Pi} \langle \phi_b - \phi_\mu, \hat{f}_t \rangle + \frac{1}{4\lambda}\|\phi_b\|_{V(\mu)^{-1}}^2 + \lambda\epsilon^2$$

*Moreover, the objective is convex in $\mu \in \mathscr{P}(\Pi)$.*

The proof is a straightforward calculation provided in Appendix C.5. Note that the saddle point expression is analogous to Eq. (5), and in fact, one can linearize the inner maximization over $\mathscr{P}(\Pi)$, such that the inner saddle point becomes convex-concave. This leads to expressions equivalent to Eqs. (4) and (6), albeit the objective is no longer linear in $\mu \in \mathscr{P}(\Pi)$. We use Lemma 5 to employ the same strategy as before: As a function of $\lambda \geq 0$, solve the inner problem of the expression in Lemma 5, for example, as a convex program with $|\Pi|$ variables and $|\Pi|$ constraints (Appendix D). Then all that remains is to solve a one-dimensional optimization problem over $\lambda \in [0, \max_{f \in \mathcal{M}} \epsilon^{-2}\text{dec}_\epsilon^{ac}(f)]$. We demonstrate this approach in Appendix E to showcase the performance of E2D on simple examples.

# 4    Conclusion

We introduced ANYTIME-E2D, an algorithm based on the estimation-to-decisions framework for sequential decision-making with structured observations. The algorithm optimizes the *average-constrained* decision-making coefficient, which can be understood as a reparametrization of the corresponding offset version. The reparametrization facilitates an elegant anytime analysis and makes setting all remaining hyperparameters immediate. We demonstrate the improvement with a novel bound for linear bandits with side-observations, that is not attained by previous approaches. Lastly, we discuss how the algorithm can be implemented for finite and linear model classes. Nevertheless, much remains to be done. For example, one can expect the reference model to change very little from round to round, and therefore, it seems wasteful to solve Eq. (3) from scratch repetitively. Preferable instead would be an incremental scheme that iteratively computes updates to the sampling distribution.

## Acknowledgments and Disclosure of Funding

Johannes Kirschner gratefully acknowledges funding from the SNSF Early Postdoc.Mobility fellowship P2EZP2_199781.

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

**Algorithm 2:** Exponential Weights Algorithm (EWA) for Density Estimation

---

**Input :** Finite model class $\mathcal{M}$, data $\mathcal{D}_t = \{(y_1, \pi_1), \ldots, (y_t, \pi_t)\}$, Learning rate $\eta > 0$

**1** Define $L(f) = -\sum_{s=1}^{t} \log M_f(y_s | \pi_s)$

**2** Let $p(f) \propto \exp(-\eta L(f))$

**3** For convex $\mathcal{M}$: Return $\sum_{f \in \mathcal{M}} p(f) f$

**4** Else: Return $f \sim p(\cdot)$.

---

## A  Online Density Estimation

For any $f \in \mathcal{M}$ and $\pi \in \Pi$, we denote by $p(\cdot | \pi, f)$ the the density function of the observation distribution $M_f(\pi)$ w.r.t. a reference measure over the observation space $\mathcal{O}$. Consider a finite model class $\mathcal{M}$ and the KL divergence,

$$e_\pi I_f e_g = \mathbb{E}_{y \sim M_g(\pi)}\left[\log\left(\frac{p(y|\pi, g)}{p(y|\pi, f)}\right)\right] \tag{13}$$

In this case, the estimation error can be written as follows:

$$\text{Est}_n = \mathbb{E}\left[\sum_{t=1}^{n} e_{\pi_t} I_{\hat{f}_t} e_{f^*}\right] = \mathbb{E}\left[\sum_{t=1}^{n} \log(p(y_t|\pi_t, f^*)/p(y_t|\pi_t, \hat{f}_t))\right]$$

$$= \mathbb{E}\left[\sum_{t=1}^{n} \log\left(\frac{1}{p(y_t|\pi_t, \hat{f}_t)}\right) - \sum_{t=1}^{n} \log\left(\frac{1}{p(y_t|\pi_t, f^*)}\right)\right]$$

The last line can be understood as the *estimation regret* of the estimates $\hat{f}_1, \ldots, \hat{f}_n$ under the logarithmic loss. A classical approach to control this term is the *exponential weights algorithm* (EWA) given in Algorithm 2. For the EWA algorithm, we have the following bound.

**Lemma 6** (EWA for Online Density Estimation). *For any data stream $\{y_1, \pi_1, \ldots, y_n, \pi_n\}$ the predictions $\hat{f}_1, \ldots \hat{f}_n$ obtained via Algorithm 2 with $\eta = 1$ satisfy*

$$\text{Est}_n \leq \mathbb{E}\left[\sum_{t=1}^{n} \log\left(\frac{1}{p(y_t|\pi_t, \hat{f}_t)}\right) - \inf_{g \in \mathcal{M}} \sum_{t=1}^{n} \log\left(\frac{1}{p(y_t|\pi_t, g)}\right)\right] \leq \log(|\mathcal{M}|) \tag{14}$$

For a proof, see [Cesa-Bianchi and Lugosi, 2006, Proposition 3.1].

### A.1  Bounding the Estimation Error of Projected Regularized Least-Squares

In this section, we consider the linear model from Example 2.2. We denote by $\|\cdot\|$ the Euclidean norm. For simplicity, the observation maps $M_\pi \in \mathbb{R}^{m \times d}$ are assumed to have the same output dimension $m \in \mathbb{N}$. The observation distribution is such that $y_t = M_{\pi_t} f^* + \xi_t$, where $\xi \in \mathbb{R}^m$ is random noise such that $\mathbb{E}_t[\xi] = 0$ and $\mathbb{E}_t\left[\|\xi\|^2\right] \leq \sigma^2$. Here, $\mathbb{E}_t[\cdot] = \mathbb{E}[\cdot | \pi_1, y_1, \ldots, \pi_{t-1}, y_{t-1}, \pi_t]$ is the conditional observation in round $t$ including the decision $\pi_t$ chosen in round $t$.

We will use the quadratic divergence[4], $e_\pi I_f e_g = \frac{1}{2}\|M_\pi(g - f)\|^2$ This choice corresponds to the Gaussian KL, but we do not require that the noise distribution is Gaussian is the following. In the linear bandit model, this choice reduces to $e_\pi I_f e_g = \frac{1}{2}\langle \phi_\pi, g - f \rangle^2$.

Let $K \subset \mathbb{R}^d$ be a closed convex set. Our goal is to control the estimation regret for the projected regularized least-squares estimator,

$$\hat{f}_t = \arg\min_{f \in K} \sum_{s=1}^{t-1} \|M_{\pi_s} f - y_s\|^2 + \|f\|_{V_0}^2 = \text{Proj}_{V_t}\left(V_t^{-1} \sum_{s=1}^{t-1} M_{\pi_s}^\top y_s\right) \tag{15}$$

where $V_0$ is a positive definite matrix, $V_t = \sum_{s=1}^{t-1} M_{\pi_s}^\top M_{\pi_s} + V_0$ and $\text{Proj}_{V_t}(\cdot)$ is the orthogonal projection w.r.t. the $\|\cdot\|_{V_t}$ norm. For $K = \mathbb{R}^d$ and $V_0 = \eta \mathbf{1}_d$, this recovers the standard ridge

---

[4]We added a factor of $\frac{1}{2}$ for convenience.

regression. The projection is necessary to bound the magnitude of the squared loss, and the result will depend on an almost-surely bound on the 'observed' diameter,

$$\max_{f,g \in K} \max_{\pi \in \Pi} \|M_\pi(f - g)\| \le B$$

Recall that our goal is to bound the estimation error,

$$\text{Est}_n = \mathbb{E}\left[\sum_{t=1}^n e_{\pi_t} I_{\hat{f}_t} e_{f^*}\right] = \mathbb{E}\left[\sum_{t=1}^n \tfrac{1}{2}\|M_{\pi_t}(f^* - \hat{f}_t)\|^2\right] \tag{16}$$

We remark that one can get the following naive bound by applying Cauchy-Schwarz:

$$\sum_{t=1}^n \|M_{\pi_t}(f^* - \hat{f}_t)\|^2 \le \sum_{t=1}^n \|M_{\pi_t}\|_{V_t^{-1}}^2 \|f^* - \hat{f}_t\|_{V_t}^2 \le \mathcal{O}(d^2 \log(n)^2) \tag{17}$$

The last inequality follows from the elliptic potential lemma and standard concentration inequalities [Lattimore and Szepesvári, 2020a, Lemma 19.4 and Theorem 20.5]. However, this will lead to an additional $d$-factor in the regret that can be avoided, as we see next.

For $K = \mathbb{R}^d$, one-dimensional observations and noise bounded in the range $[-\bar{B}, \bar{B}]$, one can also directly apply [Cesa-Bianchi and Lugosi, 2006, Theorem 11.7] to get $\text{Est}_n \le \mathcal{O}(\bar{B}^2 d \log(n))$, thereby improving the naive bound by a factor $d \log(n)$. This result is obtained in a more general setting, where no assumptions, other than boundedness, are placed on the observation sequence $y_1, \ldots, y_n$. Here we refine and generalize this result in two directions: First, we allow for the more general feedback model in with multi-dimensional observations (Example 2.2). Second, we directly exploit the stochastic observation model to obtain a stronger result that does not require the observation noise to be bounded.

**Theorem 2.** *Consider the linear observation setting with additive noise and quadratic divergence $e_\pi I_f e_g = \tfrac{1}{2}\|M_\pi(g - f)\|^2$, as described at the beginning of this section. Assume that $\max_{f,g \in \mathcal{M}, \pi \in \Pi} \|M_\pi(f - g)\| \le B$ and $\mathbb{E}\big[\|\xi_t\|^2\big] \le \sigma^2$. Then*

$$\text{Est}_n \le (\sigma^2 + B^2)\mathbb{E}\left[\log\left(\frac{\det V_n}{\det V_0}\right)\right]$$

*If in addition $\|M_\pi\| \le L$ and $V_0 = \eta \mathbf{1}_d$, then $\text{Est}_n \le (\sigma^2 + B^2)\log\big(1 + \frac{nL^2}{\eta d}\big)$.*

**Remark 1.** *Note that by the [Lattimore and Szepesvári, 2020a, Theorem 19.4], $\log\left(\frac{\det V_n}{\det V_0}\right)$ can further be upper bounded by $d \log\left(\frac{traceV_0 + nL^2}{d \det(V_0)^{1/d}}\right)$, which effectively results the desired bound.*

*Proof.* The proof adapts [György et al., 2013, Theorem 19.8] to multi-dimensional observations and takes advantage of the stochastic loss function by taking the expectation.

First, define $l_t(f) = \tfrac{1}{2}\|M_{\pi_t}f - y_t\|^2$. Then, using that $\mathbb{E}_t[y_t] = M_{\pi_t}f^*$,

$$\text{Est}_n = \mathbb{E}\left[\sum_{t=1}^n \tfrac{1}{2}\|M_{\pi_t}(f^* - \hat{f}_t)\|^2\right] = \mathbb{E}\left[\sum_{t=1}^n \tfrac{1}{2}\|M_{\pi_t}\hat{f}_t - y_t\|^2 - \tfrac{1}{2}\|M_{\pi_t}f^* - y_t\|^2\right]$$

$$= \mathbb{E}\left[\sum_{t=1}^n l_t(\hat{f}_t) - l_t(f^*)\right]$$

Further, by directly generalizing [György et al., 2013, Lemma 19.7], we have that

$$l_t(\hat{f}_{t+1}) - l_t(\hat{f}_t) \le \nabla l_t(\hat{f}_t) V_t^{-1} \nabla l_t(\hat{f}_t) = (M_{\pi_t}\hat{f}_t - y_t)^\top M_{\pi_t}^\top V_t^{-1} M_{\pi_t}(M_{\pi_t}\hat{f}_t - y_t) \tag{18}$$

We now start upper bounding the estimation error,

$$
\begin{aligned}
\text{Est}_n &\overset{(i)}{\leq} \|f^*\|^2 + \mathbb{E}\left[\sum_{t=1}^{n}\big(l_t(w_t) - l_t(w_{t+1})\big)\right] \\
&\overset{(ii)}{\leq} \|f^*\|^2 + \mathbb{E}\left[\sum_{t=1}^{n}\big(\xi_t + M_{\pi_t}(f^* - \hat{f}_{t-1})\big)M_{\pi_t}V_t^{-1}M_{\pi_t}\big(\xi_t + M_{\pi_t}(f^* - \hat{f}_{t-1})\big)\right] \\
&\overset{(iii)}{=} \|f^*\|^2 + \mathbb{E}\left[\sum_{t=1}^{n}\xi_t M_{\pi_t}V_t^{-1}M_{\pi_t}\xi_t\right] + \mathbb{E}\left[\sum_{t=1}^{n}\bar{x}_t M_{\pi_t}V_t^{-1}M_{\pi_t}\bar{x}_t\right] \\
&\overset{(iv)}{\leq} \|f^*\|^2 + \mathbb{E}\left[\sum_{t=1}^{n}\lambda_{\max}(M_{\pi_t}V_t^{-1}M_{\pi_t})\|\xi_t\|^2\right] + \mathbb{E}\left[\sum_{t=1}^{n}\lambda_{\max}(M_{\pi_t}V_t^{-1}M_{\pi_t})\|\bar{x}_t\|^2)\right] \\
&\overset{(v)}{\leq} \|f^*\|^2 + (\sigma^2 + B^2)\mathbb{E}\left[\sum_{t=1}^{n}\lambda_{\max}(M_{\pi_t}V_t^{-1}M_{\pi_t})\right]
\end{aligned}
\tag{19}
$$

The inequality $(i)$ follows from [Shalev-Shwartz et al., 2012, Lemma 2.3]. For $(ii)$ we used Eq. (18). For $(iii)$ we used that $\mathbb{E}_t[\xi_t] = 0$. In $(iv)$, we introduce the maximum eigenvalue $\lambda_{\max}(A)$ for $A \in \mathbb{R}^{m \times m}$ and denote $\bar{x}_t = M_{\pi_t}(f^* - f_{t-1})$. Lastly, in $(v)$ we used that $\|\bar{x}_t\|^2 \leq B$ and $\mathbb{E}_t[\|\xi_t\|^2] \leq \sigma^2$.

We conclude the proof with basic linear algebra. Denote by $\lambda_i(A)$ the $i$-th eigenvalue of a matrix $M \in \mathbb{R}^{m \times m}$. Using the generalized matrix determinant lemma, we get

$$
\begin{aligned}
\det(V_{t-1}) &= \det(V_t - M_{\pi_t}^\top M_{\pi_t}) \\
&= \det(V_t)\det(I - M_{\pi_t}^\top V_t^{-1} M_{\pi_t}) \\
&= \det(V_t)\prod_{i=1}^{m}(1 - \lambda_i(M_{\pi_t}^\top V_t^{-1} M_{\pi_t}))
\end{aligned}
$$

Note that $\lambda_i(M_{\pi_t}^\top V_t^{-1} M_{\pi_t}) \in (0,1]$. Next, using that $\log(1-x) \leq -x$ for all $x < 1$, we get that

$$
\log\left(\frac{\det(V_{t-1})}{\det(V_t)}\right) = \sum_{i=1}^{m}\log(1 - \lambda_i(M_{\pi_t}^\top V_t^{-1} M_{\pi_t})) \leq -\sum_{i=1}^{m}\lambda_i(M_{\pi_t}^\top V_t^{-1} M_{\pi_t})
$$

Rearranging the last display, and bounding the sum by its maximum element, we get

$$
\lambda_{\max}(M_{\pi_t}^\top V_t^{-1} M_{\pi_t}) \leq \sum_{i=1}^{m}\lambda_i(M_{\pi_t}^\top V_t^{-1} M_{\pi_t}) \leq \log\left(\frac{\det(V_t)}{\det(V_{t-1})}\right)
\tag{20}
$$

The proof is concluded by combining Eqs. (19) and (20). $\qquad\square$

**Remark 2** (Expected Regret). *The beauty of Theorem 1 is that the proof uses* only *in-expectation arguments. This is unlike most previous analysis, that controls the regret via controlling tail-events, and bounds on the expected regret are then derived a-posteriori from high-probability bounds. In the context of linear bandits, Theorem 2 leads to bound on the expected regret that only requires the noise variance to be bounded, whereas most previous work relies on the stronger sub-Gaussian noise assumption [e.g. Abbasi-Yadkori et al., 2011].*

**Remark 3** (Kernel Bandits / Bayesian Optimization). *Using the standard 'kernel-trick', the analysis can further be extended to the non-parametric setting where $\mathcal{M}$ is an infinite-dimensional reproducing kernel Hilbert space (RKHS).*

# B  PAC to Regret Bounds

## B.1  Proof of Lemma 3

*Proof.* The Lagrangian for Eq. (12) is

$$
\text{pac-dec}_\epsilon^{ac,f}(f) = \min_{\lambda \geq 0}\min_{\mu \in \mathscr{P}(\Pi)}\max_{\nu \in \mathscr{P}(\mathcal{M})}\delta_f\nu - \lambda(\mu I_f\nu - \epsilon^2).
$$

Reparametrize any $\mu \in \mathscr{P}(\Pi)$ as $\bar{\mu}(p) = (1-p)e_{\pi_f^*} + p\mu_2$. We bound $\mathrm{dec}_\epsilon^{ac,f}$ by a function of pac-$\mathrm{dec}_\epsilon^{ac,f}$. Starting from Eq. (5), we have

$$
\begin{aligned}
\mathrm{dec}_\epsilon^{ac,f}(f) &= \min_{\lambda \geq 0} \min_{\mu \in \mathscr{P}(\Pi)} \max_{\nu \in \mathscr{P}(\mathcal{M})} \mu\Delta_f\nu - \lambda(\mu I_f\nu - \epsilon^2) \\
&= \min_{\lambda \geq 0} \min_{\bar{\mu} \in \mathscr{P}(\Pi)} \max_{\nu \in \mathscr{P}(\mathcal{M})} \bar{\mu}\Delta_f\nu - \lambda(\bar{\mu} I_f\nu - \epsilon^2) \\
&= \min_{\lambda \geq 0} \min_{0 \leq p \leq 1} \min_{\mu_2 \in \mathscr{P}(\Pi)} \max_{\nu \in \mathscr{P}(\mathcal{M})} \delta_f\nu + p\mu_2\Delta_f e_f - \lambda\bar{\mu} I_f\nu - \lambda\epsilon^2 \\
&\leq \min_{\lambda \geq 0} \min_{0 \leq p \leq 1} \min_{\mu_2 \in \mathscr{P}(\Pi)} \max_{\nu \in \mathscr{P}(\mathcal{M})} \delta_f\nu + p\mu_2\Delta_f e_f - \lambda p\mu_2 I_f\nu - \lambda\epsilon^2 \\
&\leq \min_{0 \leq p \leq 1} \min_{\lambda' \geq 0} \min_{\mu_2 \in \mathscr{P}(\Pi)} \max_{\nu \in \mathscr{P}(\mathcal{M})} \delta_f\nu - \lambda'(\mu_2 I_f\nu - \frac{\epsilon^2}{p}) + p\Delta_{\max} \\
&\leq \min_{0 \leq p \leq 1} \text{pac-dec}_{\frac{\epsilon}{\sqrt{p}}}^{ac,f}(f) + p\Delta_{\max}.
\end{aligned}
$$

$\square$

## B.2   Proof of Lemma 4

*Proof of Lemma 4.* For the first part, note that

$$
\begin{aligned}
\text{pac-dec}_\epsilon^{ac,f}(f) &= \min_{\mu \in \mathscr{P}(\Pi)} \min_{\lambda \geq 0} \max_{\nu \in \mathscr{P}(\mathcal{M})} \delta_f\nu - \lambda\mu I_f\nu + \lambda\epsilon^2 \\
&= \min_{\mu \in \mathscr{P}(\Pi)} \min_{\lambda \geq 0} \max_{b \in \Pi} \max_{g \in \mathcal{M}} \langle \phi_b, g \rangle - \langle \phi_{\pi_f^*}, f \rangle - \lambda\|g - f\|_{V(\mu)}^2 + \lambda\epsilon^2 \\
&\overset{(i)}{=} \min_{\mu \in \mathscr{P}(\Pi)} \min_{\lambda \geq 0} \max_{b \in \Pi} \langle \phi_b - \phi_{\pi_f^*}, f \rangle + \frac{1}{4\lambda}\|\phi_b\|_{V(\mu)^{-1}}^2 + \lambda\epsilon^2 \\
&\overset{(ii)}{\leq} \min_{\mu \in \mathscr{P}(\Pi)} \min_{\lambda \geq 0} \max_{b \in \Pi} \frac{1}{4\lambda}\|\phi_b\|_{V(\mu)^{-1}}^2 + \lambda\epsilon^2 \\
&= \min_{\mu \in \mathscr{P}(\Pi)} \max_{b \in \Pi} \epsilon\|\phi_b\|_{V(\mu)^{-1}} \\
&\overset{(iii)}{\leq} \epsilon\sqrt{d}.
\end{aligned}
$$

Equation $(i)$ follows by computing the maximizer attaining the quadratic form over $\mathcal{M} = \mathbb{R}^d$. The inequality $(ii)$ is by definition of $\pi_f^*$ and the last inequality $(iii)$ by the assumption that the reward is observed, respectively, $\phi_\pi\phi_\pi^\top \preceq M_\pi^\top M_\pi$, and the Kiefer–Wolfowitz theorem.

The second part of the statement follows by combining Lemmas 2 and 3. $\square$

## C   Coefficient Relations Results and Proofs

**Lemma 7.** *Assume Assumption 1 holds, i.e.*

$$(\mu(r_f - r_g))^2 \leq \mu I_f e_g. \tag{21}$$

*Then*

$$\left(\sum_g \mu(r_f - r_g)\nu_g\right)^2 \leq \sum_g \left(\mu(r_f - r_g)\right)^2\nu_g \leq \mu I_f\nu.$$

*Proof.* First Jensen's inequality, then Eq. (21). $\square$

## C.1 Proof of Lemma 1

*Proof of Lemma 1.* Note that

$$
\begin{aligned}
\mathrm{dec}_\epsilon^{ac}(f) &= \min_{\mu\in\mathcal{P}(\Pi)} \max_{\nu\in\mathcal{P}(\mathcal{M})} \mu\Delta\nu \qquad \text{s.t.} \qquad \mu I_f\nu \le \epsilon^2 \\
&= \min_{\mu\in\mathcal{P}(\Pi)} \max_{\nu\in\mathcal{P}(\mathcal{M})} \mu\Delta_f\nu + \sum_{g\in\mathcal{M}}\sum_{\pi\in\Pi} \nu_g\mu_\pi(r_f(\pi)-r_g(\pi)) \qquad \text{s.t.} \qquad \mu I_f\nu \le \epsilon^2 \\
&\le \min_{\mu\in\mathcal{P}(\Pi)} \max_{\nu\in\mathcal{P}(\mathcal{M})} \mu\Delta_f\nu + \sqrt{\mu I_f\nu} \qquad \text{s.t.} \qquad \mu I_f\nu \le \epsilon^2 \\
&\le \epsilon + \min_{\mu\in\mathcal{P}(\Pi)} \max_{\nu\in\mathcal{P}(\mathcal{M})} \mu\Delta_f\nu \qquad \text{s.t.} \qquad \mu I_f\nu \le \epsilon^2 \\
&\le \epsilon + \mathrm{dec}_\epsilon^{ac,f}(f) \qquad \text{s.t.} \qquad \mu I_f\nu \le \epsilon^2,
\end{aligned}
$$

where the first inequality is by Lemma 7. Also, by lower bounding the sum in the second inequality by $-\sqrt{\mu I_f\nu}$ we get left inequality. $\qquad\square$

## C.2 Proof of Lemma 2

*Proof of Lemma 2.* For the first inequality, using the definition of $\mathrm{dec}_\epsilon^{ac}(f,\Delta_f)$ and the AM-GM inequality:

$$
\begin{aligned}
\mathrm{dec}_\epsilon^{ac,f}(f) &= \min_{\lambda\ge 0} \max_{\nu\in\mathscr{P}(\mathcal{M})} \min_{\mu\in\mathscr{P}(\Pi)} \mu\Delta_f\nu - \lambda\mu I_f\nu + \lambda\epsilon^2 \\
&\le \min_{\lambda>0} \max_{\nu\in\mathscr{P}(\mathcal{M})} \min_{\mu\in\mathscr{P}(\Pi)} \frac{(\mu\Delta_f\nu)^2}{4\lambda\mu I_f\nu} + \lambda\epsilon^2 \qquad\qquad (22) \\
&= \min_{\lambda>0} \frac{\Psi(f)}{4\lambda} + \lambda\epsilon^2 = \epsilon\sqrt{\Psi(f)}. \qquad\qquad\qquad\quad (23)
\end{aligned}
$$

Further, by Eq. (11) and Assumption 1 we have $\mu_\nu^{\mathrm{TS}}\Delta_f\nu \le \sqrt{K\sum_{g,h\in\mathcal{M}}\nu_g\nu_h e_{\pi_h^*}(r_g-r_f)^2} \le \sqrt{K\mu_f^{\mathrm{TS}}I_f\nu}$, which gives $\Psi(f)\le K$. Plugging this into Eq. (23) gives the second inequality. $\quad\square$

## C.3 Regret bound for Algorithm 1 defined for $\Delta_f$ and $\mathrm{dec}_\epsilon^{ac,f}$

**Lemma 8.** *If Assumption 1 holds, then the regret of* ANYTIME-E2D *(Algorithm 1) with $\Delta$ replaced with $\Delta_f$ is bounded as follows:*

$$
R_n \le \max_{t\in[n],f\in\mathcal{M}} \left\{ \frac{\mathrm{dec}_{\epsilon_t}^{ac,f}(f)}{\epsilon_t^2} \right\} \left( \sum_{t=1}^n \epsilon_t^2 + \mathrm{Est}_n \right) + \sqrt{n\mathrm{Est}_n}
$$

*Proof.* The proof follows along the lines of the proof of Theorem 1. The main difference is that when introducing $\Delta_f$, we get a term that captures the reward estimation error:

$$
R_n \le \sum_{t=1}^n \mathbb{E}\left[ \mathrm{dec}_{\epsilon_t}^{ac}(\hat{f}_t) + \lambda_t(\mu_t I_{\hat{f}_t} e_{f^*} - \epsilon_t^2) \right] + \sum_{t=1}^n \mathbb{E}\left[ \mu_t(r_{\hat{f}_t} - r_{f^*}) \right] \qquad (24)
$$

$$
\le \max_{t\in[n]} \max_{f\in\mathcal{M}} \left\{ \frac{1}{\epsilon_t^2}\mathrm{dec}_{\epsilon_t}^{ac}(f) \right\} \sum_{t=1}^n \left( \epsilon_t^2 + \mathbb{E}\left[ \mu_t I_{\hat{f}_t} e_{f^*} \right] \right) + \sqrt{n\mathrm{Est}_n} \qquad (25)
$$

For the last inequality, we used Cauchy-Schwarz and Assumption 1 to bound the error term,

$$
\sum_{t=1}^n \mathbb{E}\left[ \mu_t(r_{\hat{f}_t} - r_{f^*}) \right] \le \sqrt{n\sum_{t=1}^n \mathbb{E}\left[ (\mu_t(r_{\hat{f}_t} - r_{f^*}))^2 \right]} \le \sqrt{n\sum_{t=1}^n \mathbb{E}\left[ \mu_t I_{\hat{f}_t} e_f \right]} = \sqrt{n\mathrm{Est}_n}
$$

$\square$

## C.4 Generalized Information Ratio

The generalized information ratio [Lattimore and Gyorgy, 2021] for $\mu \in \mathscr{P}(\Pi)$, $\nu \in \mathscr{P}(\mathcal{M})$, and $\alpha > 1$ is defined as

$$\Psi_{\alpha,f}(\mu,\nu) = \frac{(\mu\Delta_f\nu)^\alpha}{\mu I_f \nu} \tag{26}$$

For $\alpha = 2$, we get the standard information ratio introduced by Russo and Van Roy [2014] with $\nu$ as a prior over the model class $\mathcal{M}$. Define $\Psi_\alpha(f) = \max_{\nu \in \mathcal{M}} \min_{\mu \in \mathscr{P}(\Pi)} \Psi_{\alpha,f}(\mu,\nu)$. To upper bound $\text{dec}_\epsilon^{ac}$, we have the following lemma.

**Lemma 9.** *For the reference model $f$, the ac-dec can be upper bounded as*

$$\text{dec}_\epsilon^{ac}(f) \leq \min_{\lambda > 0} \left\{ \lambda^{\frac{1}{1-\alpha}} \alpha^{\frac{\alpha}{1-\alpha}} (\alpha - 1) \Psi_\alpha(f)^{\frac{1}{\alpha-1}} + \lambda\epsilon^2 \right\} \tag{27}$$

*for $\alpha > 1$.*

*Proof.* We start by noting that for $x_1, \ldots, x_\alpha \geq 0$, from AM-GM we have that $\alpha(x_1 \cdot x_2 \cdots x_\alpha)^{1/\alpha} \leq x_1 + \cdots + x_\alpha$. Substituting $x_2 = x_3 = \cdots x_\alpha$, we get $\alpha \cdot x_1^{\frac{1}{\alpha}} \cdot x_2^{\frac{\alpha-1}{\alpha}} - x_1 \leq (\alpha - 1)x_2$. Writing $x_1 = \lambda\mu I_f\nu$ and $x_2 = \alpha^{\frac{\alpha}{1-\alpha}} \left( \frac{(\mu\Delta\nu)^\alpha}{\lambda\mu I_f\nu} \right)^{\frac{1}{\alpha-1}}$ and using the previous inequality with the $\text{dec}_\epsilon^{ac}$ program gives the result. $\square$

The information ratio $\Psi_{\alpha,f}(\mu,\nu)$ can be thought of as the Bayesian information ratio in [Russo and Van Roy, 2014] where the expectation is taken over the distribution $\nu$ of possible environments. However, for information gain $I_f$, [Russo and Van Roy, 2014] use entropy difference in the posterior distribution of $\pi_f^*$ before and after the observation is revealed.

## C.5 Proof of Lemma 5

*Proof of Lemma 5.*

$$\text{dec}_\epsilon^{ac,f}(\hat{f}_t) = \min_{\mu \in \mathscr{P}(\Pi)} \min_{\lambda \geq 0} \max_{b \in \Pi} \max_{g \in \mathcal{M}} \langle \phi_b, g \rangle - \langle \phi_\mu, \hat{f}_t \rangle - \lambda\|g - \hat{f}_t\|_{V(\mu)}^2 + \lambda\epsilon^2$$

$$= \min_{\lambda \geq 0} \min_{\mu \in \mathscr{P}(\Pi)} \max_{b \in \Pi} \langle \phi_b - \phi_\mu, \hat{f}_t \rangle + \frac{1}{4\lambda}\|\phi_b\|_{V(\mu)^{-1}}^2 + \lambda\epsilon^2 . \tag{28}$$

The first equality is by definition, and the second equality follows from solving the quadratic maximization over $g \in \mathcal{M} = \mathbb{R}^d$. To show that the problem is convex in $\mu$, note that taking inverses of positive semi-definite matrices $X, Y$ is a convex function, i.e. $((1 - \eta)X + \eta Y)^{-1} \preceq (1 - \eta)X^{-1} + \eta Y^{-1}$. In particular, $V((1 - \eta)\mu_1 + \eta\mu_2)^{-1} \preceq (1 - \eta)V(\mu_1)^{-1} + \eta V(\mu_2)^{-1}$. With this the claim follows. $\square$

# D  Convex Program for Fixed $\lambda$

Take Eq. (28) and fix $\lambda > 0$. Then we have the following saddle-point problem:

$$\min_{\mu \in \mathscr{P}(\Pi)} \max_{b \in \Pi} \langle \phi_b - \phi_\mu, \hat{f}_t \rangle + \frac{1}{4\lambda}\|\phi_b\|_{V(\mu)^{-1}}^2 + \lambda\epsilon^2$$

$$= \lambda\epsilon^2 + \min_{\mu \in \mathscr{P}(\Pi)} \max_{b \in \Pi} \langle \phi_b - \phi_\mu, \hat{f}_t \rangle + \frac{1}{4\lambda}\|\phi_b\|_{V(\mu)^{-1}}^2$$

Up to the constant additive term, this saddle point problem is equivalent to the following convex program

$$\min_{y \in \mathbb{R}, \mu \in \mathbb{R}^\Pi} y \quad \text{s.t.} \quad y \geq \langle \phi_b - \phi_\mu, \hat{f}_t \rangle + \frac{1}{4\lambda}\|\phi_b\|_{V(\mu)^{-1}}^2 \quad \forall b \in \Pi$$

$$\mathbf{1}\mu = 1$$

$$\mu_\pi \geq 0 \quad \forall \pi$$

# E Experiments

All experiments below were run on a semi-bandit problem with a "revealing action", as alluded to in the paragraph below Lemma 4. Specifically, we assume a semi-bandit model where $\mathcal{M} = \mathbb{R}^d$ and the features are $\phi_\pi \in \mathbb{R}^d$. For an instance $f^* \in \mathcal{M}$, the reward function is $r_{f^*} = \langle \phi_\pi, f^* \rangle$ for all $\pi \in \Pi$. There is one revealing (sub-optimal) action $\hat{\pi} \neq \pi_{f^*}^*$. The observation for any action $\pi \neq \hat{\pi}$ is

$$M_{f^*}(\pi) = \mathcal{N}(\langle \phi_\pi, f^* \rangle, 1) \tag{29}$$

Define $M_{\hat{\pi}} = [\phi_{\pi_1}, \ldots, \phi_{\pi_{|\Pi|}}]^\top$. Then the observation for action $\hat{\pi}$ is

$$M_{f^*}(\hat{\pi}) = \mathcal{N}(M_{\hat{\pi}} f^*, \mathbf{1}_d) \tag{30}$$

Thus, the information for any action $\pi \neq \hat{\pi}$ is

$$I_f(g, \pi) = \frac{\sigma^2}{2} \langle \phi_\pi, g - f \rangle^2 \tag{31}$$

while the information for action $\hat{\pi}$ is

$$I_f(g, \hat{\pi}) = \frac{\sigma^2}{2} \|M_{\hat{\pi}}(g - f)\|^2 = \frac{\sigma^2}{2} \sum_\pi \langle \phi_\pi, g - f \rangle^2 \tag{32}$$

For this setting $\text{Est}_n \leq \mathcal{O}(d \log(n))$ (see Appendix A.1).

## E.1 Experimental Setup

Our main objective is to compare our algorithm ANYTIME-E2D to the fixed-horizon E2D algorithm by Foster et al. [2021]. ANYTIME-E2D and E2D were implemented by using the procedure described in Section 3.4. Both ANYTIME-E2D and E2D need to solve the inner convex problem in Lemma 5. To do so we use Frank-Wolfe [Frank and Wolfe, 1956, Dunn and Harshbarger, 1978, Jaggi, 2013] for 100 steps and warm-starting the optimization at the solution from the previous round, $\mu_{t-1}$. For ANYTIME-E2D we further perform a grid search over $\lambda \in [0, \max_{g \in \mathcal{M}} \epsilon^{-2} \text{dec}^{ac,f}(g)]$ (with a discretization of 50 points) to optimize over lambda within each iteration of Frank-Wolfe. For both the E2D and ANYTIME-E2D algorithm we used the version with the gaps $\Delta$ replaced with $\Delta_f$, since we noticed that both algorithms performed better with $\Delta_f$. For E2D, the scale hyperparameter $\lambda$ was set using $\lambda = \sqrt{\frac{n}{4 \log(n)}}$ as mentioned in Foster et al. [2021, Section 6.1.1]. While for ANYTIME-E2D we set the hyper-parameter $\epsilon_t^2 = d/t$. Further, we compare to standard bandit algorithms: Upper Confidence Bound (UCB) and Thompson Sampling (TS) [Lattimore and Szepesvári, 2020a].

## E.2 Experiment 1

In this experiment, we aim to demonstrate the advantage of having an anytime algorithm. Specifically, we tune $\lambda$ in the E2D algorithm for different horizons $n = 200, 500, 1000, 2000$, but run it for a fixed horizon of $n = 2000$. As such, we expect our algorithm ANYTIME-E2D to perform better than E2D when $\lambda$ was tuned for the incorrect horizons (i.e. $n = 200, 500, 1000$). The feature dimension is $d = 3$. The number of decisions is $|\Pi| = 10$. We generated the features $\phi_\pi$ for each $\pi \in \Pi$ and parameter $f^* \in \mathbb{R}^d$ randomly at the beginning and then kept them fixed throughout the experimentation. 100 independent runs were performed for each algorithm.

The results of the experiment can be seen as the left plot in Fig. 1. As expected, our algorithm ANYTIME-E2D performs better than E2D (for $n = 200, 500, 1000$). This indicates that the E2D algorithm is sensitive to different settings of $\lambda$, which is problematic when the horizon is not known beforehand. Whereas our ANYTIME-E2D algorithm performs well even when the horizon is not known.

## E.3 Experiment 2

In this experiment, we investigate the case when $n < d^4$. As pointed out below Lemma 3, we expect improvement in this regime as the regret bound of our algorithm is $R_n \leq \min\{d\sqrt{n}, d^{1/3}n^{2/3}\}$,

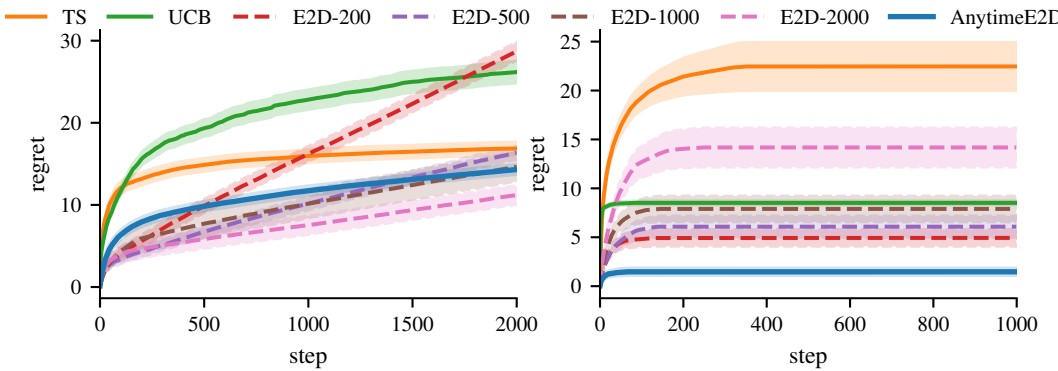

Figure 1: Running ANYTIME-E2D, TS, UCB, and E2D optimized for different horizons $n \in \{200, 500, 1000, 2000\}$. Left: The result for horizon $n = 2000$, and the feature space dimension $d = 3$. Right: The result for horizon $n = 1000$, and the feature space dimension $d = 30$.

while the default, fixed-horizon E2D algorithm cannot achieve these bounds simultaneously and one has to pick one of $d\sqrt{n}$ or $d^{1/3}n^{2/3}$ beforehand for setting the scale hyperparameter $\lambda$. It is standard that the choice of $\lambda$ is made according to the $d\sqrt{n}$ regret bound for E2D Foster et al. [2021](which is not optimal when $n \ll d^4$), especially, if the horizon is not known beforehand. Thus, we set the horizon to $n = 1000$ and the dimension of the feature space to $d = 30$, which gives us that $n = 1000 \ll 810000 = d^4$. The rest of the setup and parameters are the same as in the previous experiment except for the features $\phi_\pi$ and $f^*$ which are again chosen randomly in the beginning and then kept fixed throughout the experiment.

The results of the experiment can be seen as the right plot in Fig. 1. As expected, our algorithm ANYTIME-E2D performs better than E2D, UCB, and TS. This indicates that indeed, ANYTIME-E2D is likely setting $\lambda$ appropriately to achieve the preferred $d^{1/3}n^{2/3}$ regret rate for small horizons. The poor performance of the other algorithms can be justified, since E2D is optimized based on the worse $d\sqrt{n}$ regret rate (for small horizons), while the UCB and TS algorithms are not known to get regret better than $d\sqrt{n}$.

