# OpenReview forum: "Regret Minimization via Saddle Point Optimization"
_NeurIPS.cc/2023/Conference — NeurIPS 2023 poster_

### Official Review · Reviewer_frUC · 2023-07-04

**Soundness:** 3 good
**Presentation:** 4 excellent
**Contribution:** 2 fair
**Rating:** 4
**Confidence:** 5

**Summary:**

This paper focuses on regret minimization in sequential decision-making under uncertainty. It introduces the average-constrained decision-estimation coefficient, a saddle-point objective that characterizes the worst-case regret, enabling optimization of the information trade-off directly by the algorithm. Moreover, the paper presents a version of the Estimation-To-Decisions (E2D) algorithm (ANYTIME-E2D), with practical implementation details, improved bounds for high-dimensional linear bandits, and the first empirical results of the E2D algorithm.


**Strengths:**

The paper offers several results. The regret results are new since they are based on (3).

**Weaknesses:**

It is unclear how innovative the paper is. The brunt of the mathematics in the paper is drawn from existing prior results. Many mathematical derivations rely on standard Lagrangian (equations (3)-(7) and the proof of Lemma 1).

**Questions:**

What is the importance and impact of the new algorithm? Why is it better, more effective?

---

> ### Author Rebuttal · Authors · 2023-08-09
>
> Thank you for the review. We provide clarifications on the contributions and impact below.
>
> **Importance and impact of the new algorithm:**
> - Anytime algorithms are of great importance in practice as the horizon is often not known in advance. A theoretically sound anytime version of the E2D algorithm has not been proposed before as far as we know.
> - The anytime analysis and bounds lead to improved regret in linear bandits models in the regime $n < d^2$ (resp up to $n < d^4$ in linear feedback models), which is relevant in high-dimensional and kernel bandits - we are not aware of a similar result in the literature (Remark 2). Further note that the E2D algorithm by Foster et al (2021) uses a fixed trade-off parameter, therefore cannot achieve the optimal regret rate on different horizons simultaneously (and we are not aware of any other approach that achieves this)
> - The algorithm is different from the prior work in that the trade-off parameter is chosen for the current estimated model $\hat f_t$; The suggested choice by Foster et al (2021) is optimized for the worst-case, which, as we show, can lead to suboptimal performance, *which we  demonstrate numerically in Appendix E*.
>
>
> **Technical innovations of the paper:**
>
> While our results build on the framework by Foster et al (2021; 2023) to which we give full credit, we make several important and practically relevant contributions:
> - The anytime analysis and bounds are novel. The anytime upper bound deviates from the proof by Foster et al (2021) in a non-obvious way.
> - We explicitly derive the E2D objective for linear bandits, avoiding a computational dependence on the (infinite) model space (Lemma 6) - this has not been stated before and is important to note in particular when comparing E2D to established approaches such as UCB.
> - We prove bounds on the estimation error for regularized least squares in general linear feedback models (Theorem 2). We believe that it is important to highlight that a standard least squares estimator is feasible as an estimation oracle in the E2D framework.
> - Lemma 3 and 4 clarify the relation bounds the DEC via the information ratio IDS, DC and PAC with a direct and simple argument. A consequence of this is the improved bound for linear bandits (Lemma 5).
>
> *To conclude, we strongly believe that we make several novel and valuable contributions to the E2D framework propsed by Foster et al (2021, 2023).*
>
> We will update the paper to reflect these points better.

---

> > ### Comment · Reviewer_frUC · 2023-08-20
> >
> > Thanks for providing the answers to comments. I am staying with my score since I still believe there isn't much innovative in the paper. For example, the entire derivation of Lagrangian is standard.

---

### Official Review · Reviewer_zR5j · 2023-07-05

**Soundness:** 4 excellent
**Presentation:** 4 excellent
**Contribution:** 3 good
**Rating:** 7
**Confidence:** 3

**Summary:**

The authors consider a framework to solve the bandit problem by means of the minimax problem, which has been rapidly developed in recent years.
Among them, the paper focuses on the decision-estimation coefficient (DEC).
The DEC was developed in a series of studies by Foster+ and is known to characterize the upper and lower bound of the worst-case regret in a general class of bandit problems.
The Estimation-to-Decisions (E2D) framework, an algorithm using DEC, determines action selection probabilities based on DEC.
However, the algorithm based on offset DEC, which is an existing DEC, is expected to have conservative performance because it needs to determine the tradeoff parameter based on the regret upper bound.
Furthermore, the recently developed constrained DEC does not require a tradeoff parameter, but has the problem that the strongly duality of a Lagrangian saddle point fails.
To solve these problems, the authors propose average-constrained DEC, a formulation in which the feasible region of the max player in the constrained DEC formulation is relaxed.
This eliminates the need to determine the tradeoff parameter based on the regret upper bound while maintaining the good properties of the optimization problem, resulting in an anytime algorithm.
Furthermore, by considering a variant of average-constrained DEC, the authors show that an improved rate can be obtained for high-dimensional linear bandits, and this is confirmed through actual numerical experiments.


**Strengths:**

- Overall, the paper is written in a very clear manner. In particular, it clearly summarizes the characteristics of the DECs proposed so far.
- It clearly points out the problems of existing algorithms and presents simple solutions to solve them.
- The authors devise a specific procedure for obtaining a regret upper bound not only when the model is discrete, but also when linear bandits ( possibly with side observations) are used.
- While no numerical experimental evaluation of the DEC has been done so far, the effectiveness of the DEC is demonstrated through numerical experiments in the high dimensional linear bandits.


**Weaknesses:**

- Abstract and introduction are slightly misleading, as it appears that the average-constrained DEC results can be used directly to obtain results for high-dimensional linear bandits.
- Although partial monitoring is mentioned in l120-125, the authors are actually solving the problem in a setting where rewards are also observed (or can be assumed to be included), as described in Example 1.2. The authors do not explain to what extent there is a gap with partial monitoring, and it is not clear to what extent the proposed algorithm is applicable to partial monitoring.
- As mentioned by the authors, there are other approaches to characterizing sequential decision problems by minimax problems, such as IDS and ExpByOpt. A brief explanation of why the focus is on DEC would be desirable.

Minor issues and typos:

- l31: $M_f$ should be $M_{f^*}$
- the meaning of "localization" was unclear and it would be better to explain the meaning in short.
- In footnote 3, $\max_{f \in co(\mathcal{M})}$ is $\max_{f \in \mathcal{F}}$?


**Questions:**

- While the original DEC is formulated based on the Hellinger distance, this paper uses KL divergence for simplicity. What are the advantages and disadvantages that arise from these?
- (e.g.,l169) The meaning of telescope was unclear. Can you explain the meaning of this?
- (after l244) For the part of inequality (i), shouldn't the second term of max contains $\mathrm{dec}^{ac}_{\epsilon_t}(\hat{f}_t)$? (The order and the last equation are correct.)
- (l226) Why is the assumption "$\max_f \{ \epsilon^2 \mathrm{dec}^{ac}_{\epsilon} (f)\}$ is non-decreasing in $\epsilon$" reasonable? Is there any exception?
- Foster+2022 also devised DEC in the adversarial setting. Is the same kind of reparameterization possible in adversarial setting as in the paper? Is there any exception?

---

> ### Author Rebuttal · Authors · 2023-08-09
>
> Thank you for the review and questions. We address the points raised below:
>
> **High-dimensional bandits:**
>
> - We believe that the average-constrained DEC results do allow us to directly obtain results for high-dimensional linear bandits as shown in Remark 2. We will happily improve the clarity of our explanation if further comments are provided. The existing results on E2D imply a similar bound but only for a fixed horizon, whereas the anytime algorithm is optimal simultaneously for any stopping time. *This is a non-trivial finding not implied by prior works*.
> - The main results (Theorem 1 and also Theorem 2) do not require the rewards to be observed, and therefore apply to partial monitoring (this is the same for the existing results on E2D). Example 1.2 does not require the reward to be directly observed: It is sufficient that the reward vector is in the span of the observations. We will clarify this in the updated version of the paper. Note that Assumption 1 is only used in Lemmas 2-4.
> - We focus on the stochastic setting, where DEC was shown to tightly characterize the sample complexity of regret minimization [Foster et al. 2021, 2023]. The relation to IDS is apparent in Lemma 2, i.e. IDS respectively the information ratio certifies an upper bound on the DEC objective. In other words, without changing the proof, the bounds for DEC are always as least as good as for IDS. ExpByOpt is related but for the adversarial setting. We will make sure to clarify this further in the updated version of the paper.
>
> **Minors:** Thank you for pointing out minor issues and typos - these will be fixed and clarified in the updated version of the paper.
>
>
> **Questions:**
>
> - Hellinger vs KL: The (squared) Hellinger distance was used by Foster at al to prove near-matching upper and lower bounds. To what extent and in which cases the KL can be used instead of the Hellinger distance is currently unclear to us. In the examples we consider, the KL is sufficient. Note that the choice of divergence is independent of the main contribution of the paper - proving anytime bounds.
> From a practical perspective, the main advantage of the KL is the somewhat simpler closed-form for Gaussian distributions, the upper bounds for least-squares (Theorem 2), and the fact that the exponential weights algorithm directly bounds the KL. On a technical level, the KL upper bounds the squared Hellinger distance. Consequently, bounds on the estimation error for the KL imply bounds for the estimation error for the Hellinger distance. On the other hand, using the Hellinger distance might make it hard to prove upper bounds on the DEC.
> - By telescoping we mean that the sum over instantaneous estimation errors is bounded by the estimation error EST_n - this fact is directly used in our regret upper and the analysis by Foster et al (2021), whereas Foster et al (2023) require a stronger result, which is achieved by a more complicated refinement procedure.
> - The second term in the max in (i) is obtained by using the inequality $\lambda\_t \leq \epsilon\_t^{-2} \text{dec}^{\text{ac}}\_{\epsilon\_t}(\hat f\_t)$ (Lemma 1). Plugging this upper bound cancels out the first $ \text{dec}^{\text{ac}}\_{\epsilon\_t}(\hat f\_t) $ term so that only $ \frac{1}{\epsilon\_t^2} \text{dec}^{\text{ac}}\_{\epsilon\_t}(\hat f\_t) \mu\_t I\_{\hat f\_t} e\_{f^*}$ remains.
> - The assumption was added by mistake and is not needed to derive equation (9) from Theorem 1.
> - To what extent the results apply to the adversarial setting as in (Foster et al 2022) is an interesting question for future work. Since in this case, the trade-off parameter also corresponds to the learning rate in exponential weights, achieving an anytime result might require a different proof, i.e. an anytime analysis of FTRL.
>
>
> We will update the paper to reflect these points better.

---

> > ### Comment · Reviewer_zR5j · 2023-08-15
> >
> > Thank you for your response.
> > I have checked the replies to the questions and they are all reasonable.
> >
> > > Example 1.2 does not require the reward to be directly observed: It is sufficient that the reward vector is in the span of the observations.
> >
> > The assumption in Example 2.2 is $\phi_{\pi} \phi_{\pi}^\top \preceq M_{\pi}^\top M_{\pi}$.
> > If this is related to the assumption that the reward vector is a span of observations, why not use it as an assumption in Example 2.2?
> > In the reviewer's view, it would be easier to understand for readers as it matches the definition of observability used in partial monitoring.

---

> > > ### Author Response · Authors · 2023-08-17
> > >
> > > We do agree that introducing observability conditions would be ideal. For the current work, we focused on a simple condition that allows us to related the DEC to the decoupling coefficient and the PAC-DEC.
> > >
> > > There are trivial ways to relax the condition, e.g. by bounding the ratio$\\|\phi_\pi \phi_\pi^\top\\|/\\|M_\pi^\top M_{\pi}\\|$ in the spectral norm.
> > >
> > > It is less clear if existing analysis in finite/linear partial monitoring imply bounds on the DEC. There are subtle technical differences on how the information ratio is defined in the frequentist setting, which prevent us from directly applying the existing results, e.g. [1]. Currently, we do not know if this is just a technical obstacle, or some deeper insight is required.
> > >
> > > [1] Kirschner, Johannes, Tor Lattimore, and Andreas Krause. "Linear Partial Monitoring for Sequential Decision-Making: Algorithms, Regret Bounds and Applications." arXiv preprint arXiv:2302.03683 (2023).

---

> > > > ### Comment · Reviewer_zR5j · 2023-08-18
> > > >
> > > > I thank the authors for their responses.
> > > > I have checked the answers and they seem to be valid.
> > > > The score will remain the same as present.

---

### Official Review · Reviewer_j2sL · 2023-07-07

**Soundness:** 3 good
**Presentation:** 3 good
**Contribution:** 3 good
**Rating:** 6
**Confidence:** 2

**Summary:**

This paper studies the estimation-to-decisions framework for sequential decision-making problems with structured observations. They propose the ANYTIME-E2D algorithm, improving precious approaches with a novel bound for linear bandits. Numerical simulations are presented to show the performance of their algorithm against the baselines.

**Strengths:**

- This paper is in general well-written and the overall structure is nicely organized. The proofs seem to be theoretically rigorous.
- The significance of the paper is relatively high, since the proposed algorithm can be applied to linear bandit settings and is achieving improved rates in the high-dimensional regime, which is getting increasing attention in the literature.

**Weaknesses:**

- My first concern is the originality and significance of this work, over the existing E2D papers. It is claimed that they introduce the new objective $\operatorname{dec}_\epsilon^{a c}$, which enables the optimization of the information trade-off directly instead of the regret bound. However, no clear evidence is presented to validate the superiority of the new objective. I wonder if there are any special cases where the original objective fails to balance the trade-off, while the new one could.

- Secondly, the authors could provide more compelling motivation for the notion of 'anytime' algorithms. As a new reader of the E2D literature, I don't get the advantage of an anytime algorithm over its original version.

**Questions:**

As is discussed in the Weakness part, I hope the authors could provide more explanations and examples about the advantage of the proposed new objective and the anytime variant algorithm.

**Limitations:**

I don't see any limitations or potential negative societal impact of this work.

---

> ### Author Rebuttal · Authors · 2023-08-09
>
> Thank you for the review and valuable comments. We address the points raised below:
>
> **Significance and originality compared to prior work:**
>
> The average-constrained DEC objective and the corresponding anytime analysis has several advantages compared to the approach proposed by Foster et al:
> - Our analysis leads to practically important anytime bounds not achieved by prior work. The anytime upper bound deviates from the proof by Foster et al (2021) in a non-obvious way.
> - In linear bandits, we prove improved regret in the regime $n < d^2$ (resp up to $n < d^4$ in linear feedback models). While the existing E2D approach achieves the same bound, it does so *only for the horizon fixed before running the algorithm*. The anytime algorithm achieves the optimal trade-off simultaneously for all horizons.
> - *We validate these claims numerically in Appendix E*
> - It is true, however, that the E2D approach by Foster et al (2021) achieves (near-)optimal worst-case bounds for fixed horizon and appropriately chosen trade-off parameter. In this sense, the existing E2D paper does not “fail”.
> - Another significant difference is that the trade-off parameter in the average-constrained DEC objective is chosen for the current estimate, whereas the choice suggested by Foster et al (2021) is essentially a conservative worst-case bound derived from the analysis. Choosing the trade-off parameter adaptively can lead to improved empirical performance as we show in Appendix E.
>
> *To conclude, we strongly believe that we make several novel and valuable contributions to the E2D framework propsed by Foster et al (2021, 2023).*
>
> **Motivation for anytime algorithm:**
>
> The main advantage of the anytime algorithm is that it does not require the horizon as input parameter. This is important for several reasons:
> - The horizon might simply not be known before running the algorithm, i.e. the experimenter wishes the algorithm to keep running, possibly until some external termination criterion is met.
> - Fixing the horizon as in E2D (Foster et al, 2021) leads to optimization of the trade-off parameter for the given horizon, and the performance can be suboptimal on smaller and larger horizons - as we demonstrate theoretically (Remark 1) and numerically in Appendix E.
>
> We will update the paper to reflect these points better.

---

> > ### Comment · Reviewer_j2sL · 2023-08-21
> >
> > I thank the authors for addressing my concerns. I will keep my score because the superiority of the proposed algorithm versus Foster et al (2021, 2023) is only validated in a numerical way.

---

### Official Review · Reviewer_ye2a · 2023-07-07

**Soundness:** 2 fair
**Presentation:** 2 fair
**Contribution:** 2 fair
**Rating:** 6
**Confidence:** 2

**Summary:**

This paper focuses on regret minimization in sequential decision-making through min-max optimization. The authors introduce an anytime variant of the estimation-to-decisions algorithm that utilizes the average-constrained decision-estimation coefficient. The proposed algorithm is shown to effectively balance exploration and exploitation. The algorithm achieve a slightly improved regret performance in the context of high-dimensional linear bandits.

**Strengths:**

The paper proposed anytime variant of the estimation-to-decisions algorithm and show the algorithm could potentially improve the linear bandits for a given fixed period.

**Weaknesses:**

1) A major concern is the contribution of the paper compared to the existing literature, particularly the work by Foster et al. in 2021 and 2023. The concept of average constrained DEC appears to be very similar to the (constrained) DEC introduced in Foster et al.'s papers. It is unclear what distinguishes the introduction of the average constrained DEC in this paper and what advantages it brings.

2) The min-max problem in E2D algorithm is inefficient to solve in general, where the paper needs to find the dual variable via grid search and the choice of stepsize could be an issue.

3) While the paper states that the formulation of the average DEC leads to a practical algorithm, it would be more convincing to provide numerical results to support this claim. Comparisons with Foster et al.'s work from 2021 and 2023, as well as classical methods like UCB, Thompson sampling, and Information Direct Sampling, would justify the practicality and effectiveness of the proposed algorithm.

**Questions:**

Please see the weakness.

---

> ### Author Rebuttal · Authors · 2023-08-09
>
> Thank you for the review and valuable comments. We address the points raised below:
>
> **Contributions compared to the existing literature:**
>
> We discuss the relation to the existing E2D literature in detail in Section 3.1. While our results build on the framework by Foster et al (2021; 2023), we make several non-trivial and practically relevant contributions:
> - The anytime analysis and bounds are novel. The anytime upper bound deviates from the proof by Foster et al (2021) in a non-obvious way.
> - The algorithm is different from the prior work in that the trade-off parameter is chosen for the current estimated model $\hat f_t$; The suggested choice by Foster et al (2021) is optimized for the worst-case, which, as we show, can lead to suboptimal performance (see Appendix E for numerical results).
> - The anytime analysis and bounds lead to improved regret in linear bandits models in the regime $n < d^2$ (resp. up to $n < d^4$ in linear feedback models), which is relevant in high-dimensional and kernel bandits - and we are not aware of a similar result in the literature (Remark 2). Note further that the E2D algorithm by Foster et al (2021) uses a fixed trade-off parameter, therefore cannot achieve the optimal regret rate on different horizons simultaneously (again - we are not aware of any other approach that achieves this).
> - We explicitly derive the E2D objective for linear bandits, avoiding a computational dependence on the (continuous) model space (Lemma 6). This is important because so far it was left unclear whether E2D can be implemented even in simple case like linear bandits.
> - *We implement our approach, the E2D baseline and UCB algorithm in Appendix E* - these are the first empirical results for E2D that we are aware of.
> - We prove bounds on the estimation error for regularized least squares in linear feedback models (Theorem 2). We believe that it is important to highlight that a standard least squares estimator is feasible as an estimation oracle in the E2D framework.
>
> *To conclude, we strongly believe that we make several novel and valuable contributions to the E2D framework propsed by Foster et al (2021, 2023).*
>
> **Solving the minimax problem**
>
> - It is correct that solving the minimax problem requires a one-dimensional line search (which we show is feasible at least in smaller examples, see Appendix E). It is currently unclear whether exactly solving the minimax problem is efficient.
>
> **Numerical results**
>
> - While our focus is on the anytime analysis, we do provide numerical results in Appendix E (including E2D from Foster et al (2021), Thompson Sampling and UCB).
>
> We will update the submission to reflect these points better.
>
> We further noted that this review gives low scores for presentation and soundness. In addition to the contributions highlighted above, we kindly ask the reviewer to clarify further issues with the presentation and soundness of the paper. We will be happy to provide further clarification.

---

> > ### Comment · Reviewer_ye2a · 2023-08-16
> >
> > I want to thank the authors' response. As the paper argued that anytime E2D is the main contribution of "the anytime upper bound deviates from the proof by Foster et al (2021) in a non-obvious way." Can the authors discuss in detail the technical challenges with "replacing T with t" and how this paper addresses them? As acknowledged in the response, it seems we can just use the "anytime argument" in Foster et al (2021) to establish similar results.

---

> > > ### Author Response · Authors · 2023-08-17
> > >
> > > > As acknowledged in the response, it seems we can just use the "anytime argument" in Foster et al (2021) to establish similar results.
> > >
> > > Foster et al (2021) do **not** present an anytime analysis; nor do we say so in our response. What is true, however, is that for any **fixed** horizon, both algorithms recover the same results. The point is, however, to achieve optimal regret uniformly over time, which matters, as we show and perhaps surprisingly, even in (high-dimensional/kernel) linear bandits.
> > >
> > > > Can the authors discuss in detail the technical challenges with "replacing T with t" and how this paper addresses them?
> > >
> > > The main insight is to use the $\epsilon$-parameterized average-constrained DEC in the analysis. The technical steps are as follows: 1) re-parameterize the saddle point problem in terms of $\epsilon$, 2) Use Sion's theorem to swap the inner min/max 3) modifying the regret upper bound to include the new average-constrained quantity (which perhaps is a more natural quantity to analyze than the offset DEC; while being computationally more tractable than the hard-constrained DEC from Foster et al (2023)). 4) Use Lemma 1 to bound $\lambda^*$ in terms of the average-constrained DEC.
> > >
> > > This result is not directly implied by the existing work. Second, there are several more contributions in the paper (efficient computation for linear bandits, new bounds on estimation error, experiments, improved bounds for high-dimensional linear bandits) as we have already detailed above.

---

> > > > ### Comment · Reviewer_ye2a · 2023-08-17
> > > >
> > > > I'd like to thank the authors' response, which addresses my confusion and most of my concerns. I increase my rate.

---

### Author Rebuttal · Authors · 2023-08-09

We would like to thank all reviewers for their time and valuable inputs.
We respond to each review individually below.

We further point out that our submission contains an experimental evaluation of the proposed approach on two simple test cases in Appendix E. While the focus of the paper is on deriving the anytime bound, the experimental results include a direct comparison of E2D, TS and UCB and corroborate the claims made in Lemma 5 and Remark 2.

---

### Decision · Program_Chairs · 2023-09-21

**Decision:**

Accept (poster)

**Comment:**

The reviewers with detailed evaluations largely agreed in the opinion that this paper has strong contribution in online decision making problems. On the other hand, main concerns are the technical novelty and the relation with previous work such as Foster et al. and other approaches based on optimization like information-directed sampling. They seem to be largely solved by the rebuttals and I strongly expect that the authors carefully address these points in the final version.